# Enhanced mechanosensing of cells in synthetic 3D matrix with controlled biophysical dynamics

Boguang Yang[1,11], Kongchang Wei [1,2,11], Claudia Loebel [3], Kunyu Zhang [1,4], Qian Feng[1,5], Rui Li[1], Siu Hong Dexter Wong [1,6], Xiayi Xu[1], Chunhon Lau[7], Xiaoyu Chen[1,8], Pengchao Zhao[1], Chao Yin[1], Jason A. Burdick [3], Yi Wang [7✉] & Liming Bian [1,9,10✉]

3D culture of cells in designer biomaterial matrices provides a biomimetic cellular micro-environment and can yield critical insights into cellular behaviours not available from conventional 2D cultures. Hydrogels with dynamic properties, achieved by incorporating either degradable structural components or reversible dynamic crosslinks, enable efficient cell adaptation of the matrix and support associated cellular functions. Herein we demonstrate that given similar equilibrium binding constants, hydrogels containing dynamic crosslinks with a large dissociation rate constant enable cell force-induced network reorganization, which results in rapid stellate spreading, assembly, mechanosensing, and differentiation of encapsulated stem cells when compared to similar hydrogels containing dynamic crosslinks with a low dissociation rate constant. Furthermore, the static and precise conjugation of cell adhesive ligands to the hydrogel subnetwork connected by such fast-dissociating crosslinks is also required for ultra-rapid stellate spreading (within 18 h post-encapsulation) and enhanced mechanosensing of stem cells in 3D. This work reveals the correlation between microscopic cell behaviours and the molecular level binding kinetics in hydrogel networks. Our findings provide valuable guidance to the design and evaluation of supramolecular biomaterials with cell-adaptable properties for studying cells in 3D cultures.

[1] Department of Biomedical Engineering, The Chinese University of Hong Kong, Hong Kong, China. [2] Empa, Swiss Federal Laboratories for Materials Science and Technology, Laboratory for Biomimetic Membranes and Textiles, St. Gallen, Switzerland. [3] Department of Bioengineering, University of Pennsylvania, Philadelphia, PA, USA. [4] Department of Materials Science and Engineering, Johns Hopkins University, Baltimore, MD, USA. [5] Key Laboratory of Biorheological Science and Technology, Ministry of Education College of Bioengineering, Chongqing University, Chongqing, China. [6] Department of Biomedical Engineering, The Hong Kong Polytechnic University, HongKong, China. [7] Department of Physics, The Chinese University of Hong Kong, Hong Kong, China. [8] Department of Mechanical Engineering, Massachusetts Institute of Technology, Cambridge, MA, USA. [9] Shenzhen Research Institute, The Chinese University of Hong Kong, Hong Kong, China. [10] China Orthopedic Regenerative Medicine Group (CORMed), Hangzhou, Zhejiang, China. [11]These authors contributed equally: Boguang Yang, Kongchang Wei. ✉email: yiwang@cuhk.edu.hk; lbian@cuhk.edu.hk

I n living organisms, cells are constantly interacting with and remodeling the surrounding highly dynamic extracellular matrix (ECM), and this process enables various cell behaviors[1] including proliferation, migration, differentiation, and apoptosis[2–6]. Besides the intrinsic thermodynamic rearrangements, two orthogonal external sources, the degradation of biopolymers and the force-induced dissociation/re-association of physical crosslinks, contribute to the dynamic properties of the biopolymer network in the natural ECM. The latter typically operates at a significantly shorter timescale and higher frequency than the former, thereby giving rise to the temporal hierarchy of the dynamic behavior of the ECM[7]. The temporal hierarchy of ECM dynamics allows cells to remodel their pericellular ECM via the biosynthesis of matrix-degrading enzymes and generation of cellular forces to mediate the restructuring and rearrangement of the biopolymer network at timescales matching those of the above two cellular processes. However, the natural ECM is highly complex, and precisely manipulating its dynamic properties remains challenging. Therefore, designing a three-dimensional (3D) polymeric matrix with tunable dynamic properties to recapitulate the temporal hierarchy of ECM dynamics is of great importance for decoupling the effects of 3D matrix dynamics on cell behaviors.

Elegant prior studies have employed enzymatically degradable crosslinkers in otherwise non-degradable and static hydrogels to emulate the biochemically derived ECM dynamics[8]. These studies demonstrate that such hydrogel network dynamics based on cell-mediated matrix degradation are essential for cellular development in a 3D matrix, including cell spreading, migration, multicellular assembly, and differentiation[9–16]. However, notably, the dynamic properties of the enzymatically degradable hydrogels depend on the rate of biosynthesis of catabolic enzymes, which is affected by many factors, including cell type and culture conditions[17]. Furthermore, the network dynamics of the degradable hydrogels at a microscopic level are spatially heterogeneous and temporally inconsistent due to irreversible changes in the network[7]. These characteristics of degradable hydrogels make studying the effect of hydrogel matrix dynamics on cellular activity challenging.

To capture the intrinsic ECM dynamics derived from its physical crosslinks, extensive research effort has been dedicated to the development of dynamic cell-adaptable hydrogels crosslinked by dynamic and reversible bonds[18–30]. Unlike cell-mediated matrix degradation, the dynamic nature of adaptable hydrogels relies on intrinsically reversible crosslinks and is spatially and temporally consistent[7]. Cell-adaptable hydrogels provide a more permissive environment for encapsulated cells to interact with the hydrogel matrix. For example, Anseth et al. show that encapsulated cells can adopt a stretched morphology in polyethylene glycol (PEG) hydrogels crosslinked by dynamic hydrazone bonds[20]. Mooney et al. demonstrate that the fast stress-relaxation properties of ionically crosslinked alginate hydrogels promote mechanosensing and osteogenic differentiation of stem cells[22]. These studies all focused on how macroscopic dynamic mechanical properties of the hydrogel influenced cell behaviors. Meanwhile, the binding kinetics of the dynamic bonds fundamentally dictates the dynamic properties of the pericellular hydrogel networks and therefore should impact cellular functions in the 3D hydrogel matrix. Furthermore, the dynamic hydrogel network needs to reorganize at a matching timescale under the probing of the cellular forces to support efficient cell–matrix interactions and associated cellular behaviors such as spreading and mechanosensing[31]. Lastly, the efficient transmission of cell traction forces via the conjugated cell-adhesive ligands to the dynamic network structure is also essential to the cell-mediated adaptation of the hydrogel network.

In this study, we investigate the role of the binding kinetics of dynamic crosslinks, which determines the timescale of hydrogel network dynamics, and the conjugation stability of cell-adhesive

| Table 1 The kinetic and thermodynamic constants for complexations between cyclodextrin and adamantane or cholic acid. | | | |
|---|---|---|---|
| Host–guest | $k_{on}$ (M$^{-1}$s$^{-1}$) | $k_{off}$ (s$^{-1}$) | $K_{eq}$ (M$^{-1}$) |
| CD-ADA short lifetime | ~$10^8$ | ~$10^3$ | ~$10^5$ |
| CD-CA long lifetime | ~$10^1$ | ~$10^{-3}$ | ~$10^4$ |

Due to conjugation at the carboxylate end, the association and dissociation of CD-CA in our hydrogels can only occur through the steroid body side of CA. Kinetic constants for such association and dissociation were estimated from experimental values[55] and free energy calculations. Equilibrium constants ($K_{eq}$) were obtained by Isothermal Titration Calorimetry (see Supplementary Information for details).

ligands in cell–hydrogel network interactions and associated impact on stem cell behaviors including mechanosensing and differentiation in 3D matrix. Our experimental findings show that the combination of dynamic crosslink with short lifetime and stable conjugation of cell-adhesive ligands to hydrogel network promotes rapid stellate spreading, mechanosensing, and osteogenic differentiation of encapsulated stem cells in 3D hydrogels, and computer modeling result pinpoints the critical role of kinetic constants of dynamic crosslinks in regulating cell spreading in 3D hydrogel network. Furthermore, we prove that ultra-rapid cell spreading and assembly in the dynamic hydrogels is mediated by the concerted action of cell adhesion structures containing $\beta_1$ class integrins, interaction with nascent ECM proteins, and intracellular actomyosin contractility. Our findings highlight the importance of the careful selection of dynamic crosslinks and precise biofunctionalization to the design of dynamic cell-adaptable hydrogels for supporting rapid cellular developments in a 3D matrix.

## Results

**The network dynamics of supramolecular hydrogels is dependent on the binding kinetics of reversible dynamic crosslinks.** Based on a previously developed approach[23], we first prepared two types of physically crosslinked supramolecular hydrogels by pre-assembling mono-acryloyl cyclodextrin (ac-β-CD) on hyaluronic acid (HA) grafted with the similar amount (degree of modification ~30%) of either adamantane (HA–ADA) or cholic acid (HA–CA) via either CD–ADA or CD–CA host–guest complexation prior to photopolymerization. The obtained hydrogels, i.e., HA–ADA and HA–CA hydrogels, are solely stabilized by either CD–ADA or CD–CA host–guest complexation, respectively. These two guests (ADA and CA) were chosen because of their similar equilibrium binding constant ($K_{eq}$) but drastically different kinetic binding constants ($k_{off}$, $k_{on}$) with the host CD molecule. $K_{eq}$ of these two pairs of host–guest complexation is within an order of magnitude of each other, whereas CD–ADA possesses over six orders of magnitude larger kinetic binding constants than CD–CA (Fig. 1a and Table 1). The crosslinking density and therefore the bulk mechanical properties of host–guest hydrogels depend on $K_{eq}$, which determines the fraction of bound host–guest crosslinks, and the content of guest-functionalized polymers (kept at 4% w/v). As a result, HA–ADA and HA–CA hydrogels show similar degrees of swelling in cell culture media (Fig. 1b) and Young's modulus (1729 ± 77 and 1716 ± 71 Pa, Supplementary Fig. 3a) at swollen state.

Rheological frequency sweep test shows that both $G'$ and $G''$ of the HA–ADA hydrogels increase significantly with increasing shear frequency. In contrast, the $G'$ for HA–CA hydrogels displays a frequency-independent plateau, and $G''$ is much lower than $G'$ at the high-frequency range (Supplementary Fig. 3b, c). Previous findings suggest that the frequency of cell mechanosensing is about 0.1 Hz[32–34]. For this reason, we measured $G'$ and $G''$ of hydrogels

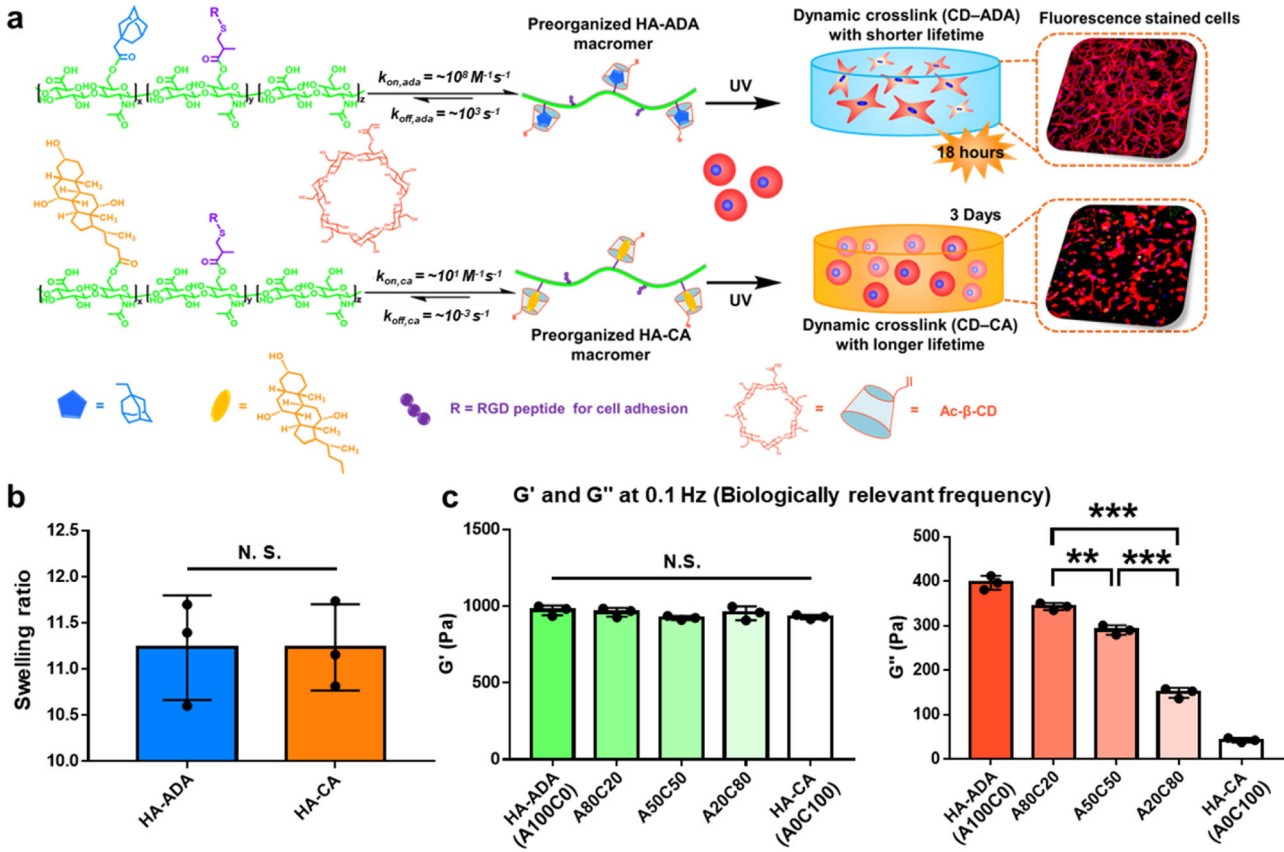

**Fig. 1 The supramolecular hydrogels stabilized by reversible host–guest crosslinks with different binding kinetics possess differential dynamic properties. a** Schematic illustration of the preparation of supramolecular hyaluronic acid hydrogels stabilized by different pairs of host–guest complexation (cyclodextrin–adamantane/cyclodextrin–cholic acid or CD–ADA/CD–CA) and the monitoring of 3D spreading of encapsulated hMSCs. **b** The swelling ratio of hydrogels measured after incubating in culture medium for 3 days. Data are presented as mean values ± SD (standard deviation), $n = 3$ independent hydrogels per group, N.S. indicates no statistical difference. (two-tailed Student's $t$-test). **c** Average value of $G'$ and $G''$ from rheological analysis at the frequency of 0.1 Hz and 1% strain. Data are presented as mean values ± SD, $n = 3$ independent hydrogels per group, N.S. indicates no statistical difference (ANOVA), $**p < 0.01$, $***p < 0.001$ (two-tailed Student's $t$-test) (A80C20, A50C50, A20C80 were prepared by mixing HA–ADA and HA–CA at the weight ratio of 80%:20%, 50%:50%, or 20%:80%, respectively).

prepared with different weight ratios of HA–ADA and HA–CA guest polymers at this biologically relevant frequency. All hydrogels possess a similar level of elasticity ($G' \sim 1$ kPa), which is close to that of biological tissues (Fig. 1c). The $G''$ of the hydrogels increases with an increasing percentage of HA–ADA guest polymer in the hydrogels (Fig. 1c), indicating an increasing level of network dynamics. Furthermore, the HA–ADA hydrogels exhibit significantly faster stress relaxation compared with the HA–CA hydrogels under compression, and the half stress-relaxation time decreases with an increasing percentage of HA–ADA guest polymer in the hydrogels (Supplementary Fig. 3d). These data suggest that increasing the fraction of CD–ADA crosslink while decreasing that of CD–CA crosslink enhances the hydrogel network dynamics. This can be attributed to the larger kinetic binding constants of CD–ADA crosslink, which are inversely related to the lifetime of dynamic bonds and therefore positively related to the hydrogel network dynamics. These findings also demonstrate the precise tuning of hydrogel network dynamics while maintaining the bulk elastic mechanical properties by adjusting the relative weight ratio of reversible supramolecular crosslinks, which possess distinct kinetic binding constants but similar equilibrium binding constants. We speculate that there exists a range of hydrogel network dynamics engendered from carefully selected dynamic crosslinks that allows for efficient rearrangement of the hydrogel network structure in response to cellular forces at a timescale that matches

that of cellular dynamics, thereby presenting the encapsulated cells with a cell-adaptable 3D matrix microenvironment.

**Dynamic crosslinks with a short lifetime promote rapid stellate spreading and mechanosensing of encapsulated cells in 3D hydrogels.** We next investigated the behaviors of hMSCs encapsulated in the supramolecular hyaluronic acid hydrogels ($1 \times 10^7$ cells/mL) stabilized by the dynamic host–guest crosslinks of different lifetimes. Covalently crosslinked methacrylated HA (MeHA) hydrogels were used as the control static hydrogels. To facilitate cell adhesion to the hydrogel matrix, RGD peptides were chemically conjugated (cRGD) to the network of all hydrogels at the same dose (Fig. 1a). After culturing in a growth medium for 2 weeks, live/dead staining and DNA quantification show that over 95% of encapsulated cells remain viable in all hydrogels (Supplementary Figs. 4 and 5). Remarkably, the spreading rates of hMSCs in the HA–CA–cRGD hydrogels (stabilized by crosslink with longer lifetime) and MeHA–cRGD (stabilized by permanent crosslink) are significantly lower compared with that in the HA–ADA–cRGD hydrogels (stabilized by crosslink with shorter lifetime) (Fig. 2a and Supplementary Figs. 4, 6). Unlike the round and non-spreading cells observed in the MeHA–cRGD and HA–CA–cRGD hydrogels, the cells in the HA–ADA–cRGD hydrogels develop extensive stellate spreading and interconnected cellular networks, which are typically observed from cells cultured

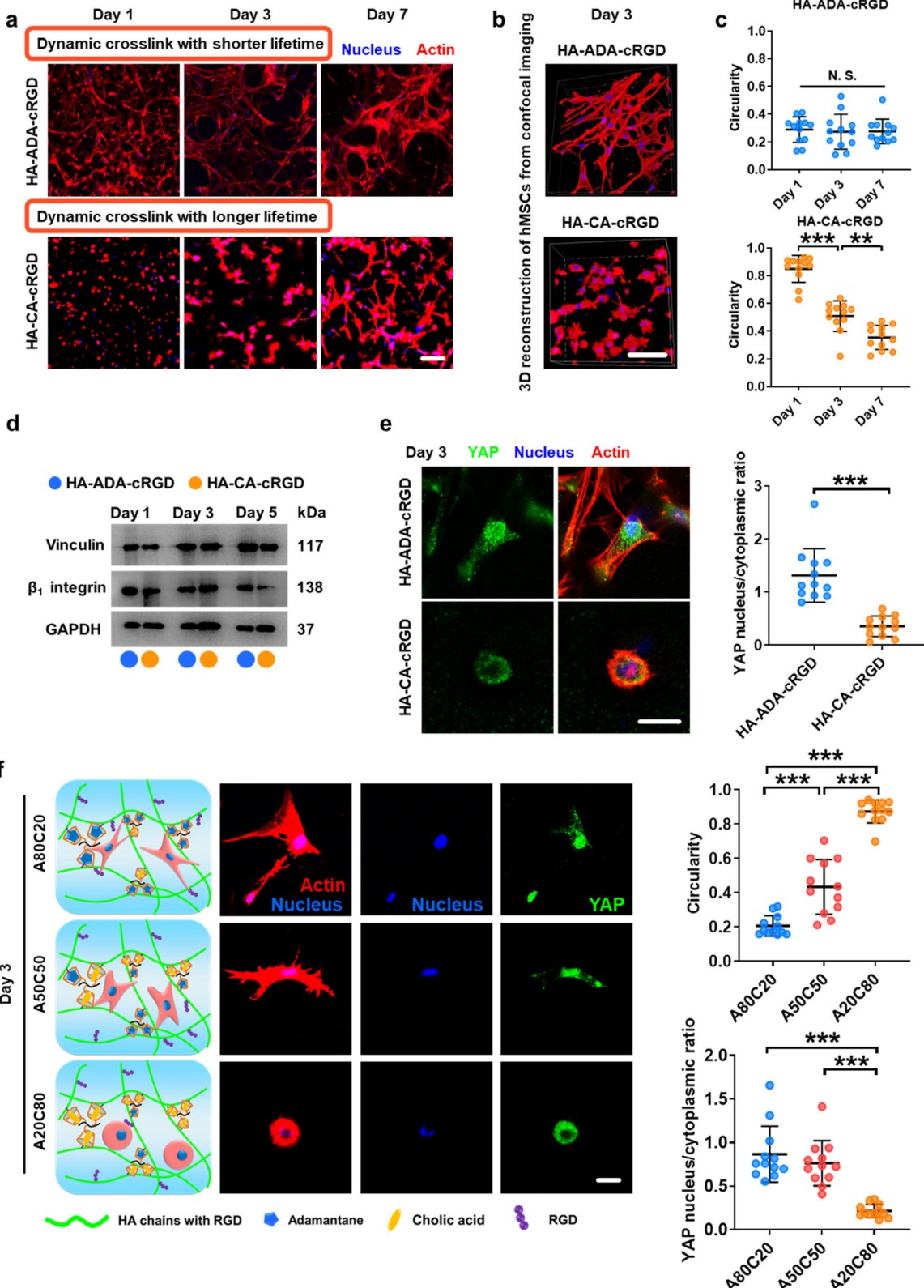

on 2D substrates, after only 18 h of culture (Fig. 2a and Supplementary Fig. 4, Supplementary Movie 1). Cells in the HA–CA–cRGD hydrogels only slowly develop the spreading morphology after 7 days of culture (Fig. 2a).

The more substantial spreading and cytoskeletal filamentous actin of cells in HA–ADA–cRGD hydrogels compared with that in HA–CA–cRGD hydrogels is further verified by 3D reconstruction of confocal microscopy imaging (Fig. 2b). The shape circularity index of cells in HA–ADA–cRGD hydrogels rapidly decreases to ~0.3 on day 1 and remains low for the rest of culture, whereas the cell circularity of HA–CA–cRGD group is significantly higher on day 1 (~0.8) and only gradually decreases to ~0.4 after 7 days

**Fig. 2 hMSCs exhibit differential spreading within 3D hydrogels stabilized by dynamic crosslinks with varying lifetimes. a** Representative image of hMSCs encapsulated in hydrogels from different groups (HA–ADA–cRGD and HA–CA–cRGD) on culture days 1, 3, and 7 stained for actin (red) and nuclei (blue). Scale bar =100 μm. **b** 3D reconstruction of hMSCs from confocal images. Scale bar =100 μm. **c** The circularity of the hMSCs encapsulated within the hydrogels from the different groups (HA–ADA–cRGD and HA–CA–cRGD) (The average cell circularity index was calculated according to $C = 4\pi A/P^2$, where $A$ is the cell area and $P$ is the cell perimeter). Data are presented as mean values ± SD, $n = 12$ cells per group from 2 independent hydrogels; N.S. indicates no statistical difference, $**p < 0.01$, $***p < 0.001$ (ANOVA or two-tailed Student's $t$-test). **d** Western blot analysis of vinculin and $\beta_1$ integrin protein expression in hMSCs after 3 days of osteogenic culture in hydrogels prepared with different lifetime crosslinks. The samples are derived from the same experiment, and gels/blots are processed in parallel. (Data of Relative intensity details are in the Supplementary Table 2) **e** Representative immunofluorescence staining against F-actin (red), nuclei (blue), and YAP (green) in hMSCs encapsulated in HA–ADA–cRGD and HA–CA–cRGD hydrogels after 3 days of culture (scale bar = 50 μm) and quantification of the nuclear YAP fluorescence intensity (intensity ratio between the nucleus and cytoplasm) Data are presented as mean values ± SD, $n = 12$ cells per group from two independent hydrogels; $***p < 0.001$ (two-tailed Student's $t$-test). **f** Representative immunofluorescence staining against F-actin (red), nuclei (blue), and YAP (green) in hMSCs encapsulated in hydrogels prepared with different weight ratios of HA–ADA and HA–CA guest polymer after 3 days of culture (A80C20, A50C50, A20C80 were prepared by mixing HA–ADA–cRGD and HA–CA–cRGD at the weight ratio of 80%:20%, 50%:50%, or 20%:80%, respectively) and quantification of the cell circularity and nuclear YAP fluorescence intensity (intensity ratio between the nucleus and cytoplasm). Data are presented as mean values ± SD, $n = 12$ cells per group from two independent hydrogels, $***p < 0.001$ (two-tailed Student's $t$-test), Scale bar = 25 μm.

(Fig. 2c). After 3 days of culture, Western blot analysis shows higher expression of vinculin and $\beta_1$-integrin in the HA–ADA–cRGD hydrogels compared with that in the HA–CA–cRGD hydrogels (Fig. 2d and Supplementary Table 2). Immunostaining shows a more significant nuclear translocation of YAP, a mechanosensing transcription factor, in the HA–ADA–cRGD hydrogels compared with that of the HA–CA–cRGD hydrogels (Fig. 2e). In the hydrogels prepared with different weight ratios of HA–ADA and HA–CA guest polymers, both cell spreading and YAP nuclear presence increase with an increasing percentage of the HA–ADA in the hydrogels (Fig. 2f). For example, A80C20 hydrogels (prepared with 80% HA–ADA–cRGD and 20% HA–CA–cRGD) have the smallest cell circularity index and highest YAP nuclei intensity among all hydrogels. Our degradation test shows that both the cell-laden HA–ADA hydrogels and HA–CA hydrogels have insignificant changes in volume over 14 days of culture, thereby indicating the good long-term stability and slow degradation of the hydrogels during this period (Supplementary Fig. 7a, b). Moreover, because the ultra-rapid stellate cell spreading in HA–ADA–cRGD hydrogels develops within less than one day, the dynamic crosslink rather than hydrogel degradation should be the primary factor contributing to the hydrogel network dynamics and resultant rapid cell spreading in 3D.

These findings together indicate that the combination of dynamic crosslinks with short lifetime (large dissociation rate constant) and static chemical conjugation of cell adhesion ligands can better facilitate the rapid and extensive spreading, development of actin cytoskeleton and cell adhesive structures, as well as mechanosensing signaling of encapsulated hMSCs in 3D hydrogels. The dynamic crosslinks with a shorter lifetime could engender sufficient hydrogel network dynamics to promote the cell-adhesive ligands clustering, which enhances cell spreading and mechanosensing[35]. More importantly, for 3D cell culture in hydrogels, the pericellular hydrogel polymeric network constitutes the 3D confinement of cells, and the dynamics of the hydrogel network dictates the "opening" frequency of this pericellular "gates" through which the dynamically assembled cell membrane protrusions (e.g. filopodia) need to pass[36]. For the hydrogel network connected by crosslinks with a long lifetime (i.e., covalent crosslink, CD–CA complexation), the "gates" can take too long to open when cells "press" against it by cellular forces and therefore remain closed to hinder cell spreading. In contrast, the pericellular hydrogel network connected by dynamic crosslinks with a short lifetime can timely adapt and reorganize in response to cellular forces and may therefore better facilitate the passing through and extension of cell protrusion structures and cell spreading. We next performed computer modeling to examine these hypotheses.

**Computer modeling pinpoints the critical role of kinetic constants of dynamic crosslinks in regulating cell spreading in a 3D hydrogel network.** Cell spreading has long been studied using the well-established stochastic "motor-clutch" model[31,34,37]. While these studies underscored the importance of substrate stress relaxation, they were performed on 2D substrates and did not explicitly model the kinetic properties of the cross-linking molecules. Here, we combined molecular dynamics (MD) and kinetic Monte Carlo (KMC) calculations to investigate the relationship between the 3D cell spreading in our hydrogels and the binding kinetics of host–guest crosslinks. These two calculations complement each other, with MD providing a detailed, atomistic picture of how cellular forces applied on a single host–guest pair may (or may not) induce hydrogel network reorganization, and KMC depicting, at a more coarse-grained level, such a picture with multiple host–guest pairs on a timescale longer than that examined by MD. In our MD calculations, we constructed an all-atom model with explicit water containing 3 HA chains interconnected by 6 CD–ADA or CD–CA crosslinks and investigated the effect of forces ranging from 10 to ~100 pN[38–40] applied along a HA chain with a given host–guest pair either in the unbound or bound state. When a given host–guest crosslink happens to be in the bound state, our MD simulations (Fig. 3a, b and Supplementary Table 3, Supplementary Fig. 9a) indicate that forces generated by the actin polymerization and F-actin extension alone, which are typically far below 100 pN[41], generally cannot unlock the bound host–guest crosslinks on the simulation timescale (5 ns). Consistent with this finding, an estimation using the Bell model (see Supplementary Information for details) confirms that even the "fast" CD–ADA pair requires approximately a millisecond to unbind under the aforementioned forces. In contrast, when a host–guest pair happens to be in the unbound state, our simulations show that increasing the magnitude of applied force results in an increased correlation between the direction of force and chain movement (i.e., force-directed HA chain movement), increased chain speed, and decreased probability of the reunion of unbound host/guest molecules with their original partners (Fig. 3c, d and Supplementary Fig. 9b). These observations indicate that when host–guest crosslinks are in the unbound state, small (≪100 pN) cellular forces can readily enable hydrogel network reorganization by displacing the free guest molecules and the segments of HA chains they decorate. According to the Bell model, the larger the base $k_{off}$ of a host–guest pair, the faster its unbinding occurs under a given applied force. Therefore, rapid dissociation kinetics (short crosslink lifetime and large $k_{off}$) is a key factor to ensure that the hydrogel network can be reorganized in time for cell spreading. To explore this scenario with multiple

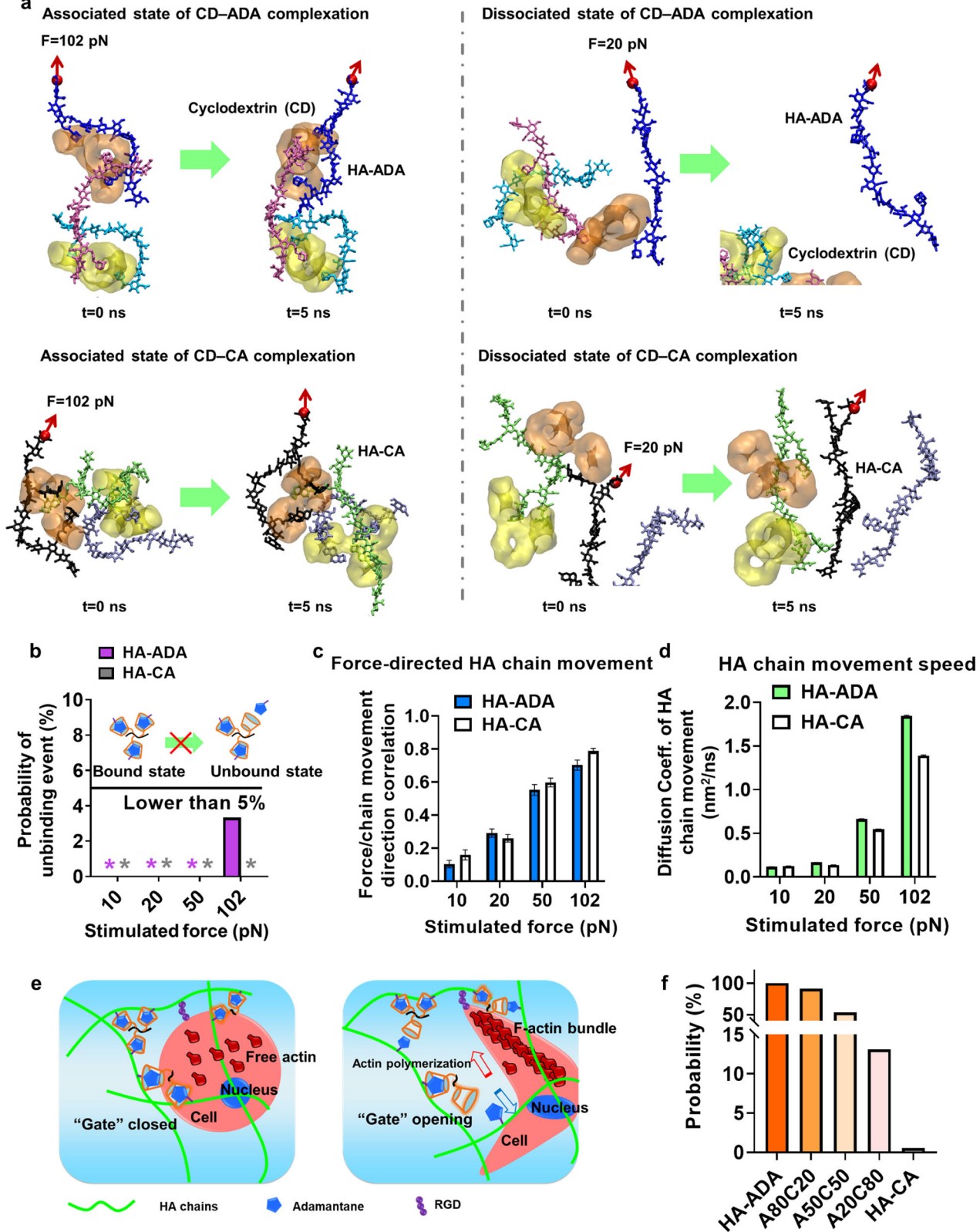

host–guest pairs on the typical timescale of actin polymerization (~0.01 s)[31,42], we turned to KMC calculations.

In KMC, we modeled the initiation of the cell spreading process as a bundle of parallel actin filaments in the protrusive filopodia of cells pushing against a network of polymers connected by the multivalent host–guest crosslinks (see Fig. 3e and Supplementary

Information for details)[43–47]. Based on our hydrogel composition, the addition of each actin monomer during the growth of a single actin filament in filopodia requires the dissociation of up to 8 host–guest crosslinks, which we term "a gate-opening event". Using kinetic constants of host–guest crosslinks and the force-dependent diffusion coefficients determined in our MD simulations, we

**Fig. 3 MD and KMC simulation results verify the key role of binding kinetics of host–guest crosslinks in enabling cell-mediated hydrogel network reorganization. a** MD simulation snapshots showing an application of a force on one end (red dot) of the HA segment when the host–guest crosslink in HA–ADA or HA–CA hydrogels is in the bound and the unbound state, respectively. Two oligomerized acryloyl β-cyclodextrin crosslinkers (each contains three CDs) are shown in orange and yellow, respectively. **b** For a given magnitude of applied force, sixty 5-ns simulations were performed with CD–ADA pairs in the bound state. Among the total 240 simulations, unbinding events were only observed in two simulations with $F = 102$ pN. * means that the value is 0. For the CD–CA pairs in the bound state, 60 simulations performed at $F = 102$ pN revealed no unbinding event. **c** In a separate set of 480 5-ns simulations with a given CD–ADA or CD–CA pair initially in the unbound state, we measured the correlation coefficient (value ranges from −1 to 1) between the direction of the applied force and the movement of the corresponding HA chain. Error bars represent the standard error of the mean. $n = 60$. **d** In the same set of simulations shown in **c**, the "speed" of HA chain movement is characterized by the diffusion coefficient of the guest molecule closest to the HA end being pulled. The sixty simulations performed at a given force are combined to estimate the diffusion coefficient from a linear regression model, the error of which is smaller than the line width (Supplementary Table 3). **e** Schematics of parallel actin bundles in filopodia and the crosslinked HA network they face. **f** Probability for a gate-opening event to occur within the estimated timescale of actin polymerization (~0.01 s) at $F = 10$ pN based on sets of 100,000 KMC calculations assuming $n = 2$ in acrylated host complexes (see Supplementary Information for details).

estimated the time it would take for such a gate-opening event to occur in KMC. Our calculations for hydrogels prepared with varying molar ratios of fast (CD–ADA) and slow (CD–CA) crosslinks (i.e., A80C20, A50C50, and A20C80) show that the probability of a "fast enough" gate-opening event, i.e., the gate opens within the typical actin polymerization timescale of ~0.01 s, increases with an increasing percentage of the fast crosslinks in the hydrogels (Fig. 3f). For example, given the same force, this probability is the highest in A80C20 among all mixed hydrogels (Fig. 3f). In addition, with the equilibrium binding constant $K_{eq}$ kept unchanged, both the required force and the average time of gate-opening decrease with increasing $k_{off}$ (Supplementary Fig. 9f). These findings are consistent with our experimental observation that cells spread faster in hydrogels with a higher percentage of fast crosslinks. Collectively, they reveal a consistent trend that supports the strong correlation between dissociation kinetics of host–guest crosslinks and cell spreading in the corresponding hydrogel network. Mechanistically, this correlation may have arisen from the need to match the timescale of hydrogel network changes (gate-opening) with that of cellular activities (actin polymerization), in line with recent findings from a 2D motor-clutch model of viscoelastic substrates[31].

**The precise and firm conjugation of cell-adhesive ligands determines the mechanosensing of encapsulated cells in dynamic hydrogels.** Mounting researches have demonstrated the importance of cell adhesion structures, which rely on the hydrogel-borne cell-adhesive ligands, in the cell-mediated hydrogel network adaptation. To better understand the mechanism of cell interactions with the hydrogel network, hMSCs were encapsulated in A50C50 hydrogels (prepared with an equal weight mixture of HA–ADA and HA–CA guest polymers). Unlike previous experiments, the cell-adhesive RGD peptides were selectively and covalently conjugated either to the fast HA–ADA subnetwork (HA–ADA–cRGD, i.e., A50-cRGD:C50) or to the slow HA–CA subnetwork (HA–CA–cRGD, i.e., A50:C50-cRGD) during hydrogel preparation (Fig. 4a). After 3 days of culture, hMSCs in the A50-cRGD:C50 hydrogels show significantly more spreading and nuclear YAP signal compared with that in the A50:C50-cRGD hydrogels (Fig. 4a, b). Because both types of hydrogels are based on the same recipe of guest polymers, they should have the same macroscopic mechanical properties and level of network dynamics. This finding indicates that only conjugating the cell-adhesive motifs to the network components connected by the more dynamic crosslinks (fast binding kinetics and short bond lifetime, i.e., CD–ADA but not CD–CA) can enable the efficient mechanosensing of encapsulated cells. The mechanosensing of cells in the 3D hydrogel network not only depends on the extent of network dynamics but also

requires precise engagement of cell adhesion apparatus with the dynamically responsive elements of the hydrogel network.

After confirming the necessity to conjugate the cell-adhesive ligands to HA–ADA guest polymer in the hydrogel network to promote cell mechanosensing, we used the RGD-grafted β-CD to physically conjugate RGD to the pure HA–ADA hydrogel matrix (HA–ADA–pRGD, 100% HA–ADA) via CD–ADA complexation[48] to compare with the hydrogels bearing covalently conjugated RGD (HA–ADA–cRGD) (Fig. 4c). Cells encapsulated within the 3D matrix of HA–ADA–pRGD hydrogels retain their round morphology after 1 day of culture (Fig. 4d and Supplementary Fig. 10, Supplementary Movie 3) and show minimal spreading even after 3 days (Supplementary Movie 4), as also observed in the hydrogels without RGD coupling (HA–ADA), whereas cells spread extensively in the HA–ADA–cRGD hydrogels as described in previous sections (Fig. 2a and Supplementary Fig. 4, Supplementary Movie 2). The lack of cell spreading both on and within HA–ADA hydrogels indicates the necessity of firm cell adhesion ligands in hydrogels to support cell spreading.

The maturation of cell adhesion structures involves the recruitment and activation of associated adaptor proteins and signaling factors, such as vinculin and focal adhesion kinase (FAK). Our immunofluorescence staining reveals intense staining of vinculin and phosphorylated FAK (pFAK) at the periphery of spreading hMSCs encapsulated in the HA–ADA–cRGD hydrogels (Fig. 4e and Supplementary Figs. 11a, 12). Consistent with the staining results, our RT-PCR analysis shows significant upregulation of vinculin gene expression after 3 days of culture in the HA–ADA–cRGD hydrogels compared with that in the control groups (HA–ADA–pRGD and HA–ADA) (Supplementary Fig. 11b). Western blot analysis reveals higher pFAK expression in hMSCs encapsulated in the HA–ADA–cRGD hydrogels compared with that in the control groups (Supplementary Fig. 13 and Supplementary Table 4). Our immunofluorescence staining also shows the substantial nuclear translocation of YAP in hMSCs encapsulated in the HA–ADA–cRGD hydrogels after 3 days of culture (Figs. 2e and 4f), whereas only a diffuse cytoplasmic YAP presence is found in cells cultured in the control hydrogels (HA–ADA–pRGD and HA–ADA) (Fig. 4f and Supplementary Fig. 14). These findings show that the precise and firm conjugation of cell-adhesive ligands to the hydrogel subnetwork connected by host–guest crosslinks with fast binding kinetics is required to support efficient cell spreading, adaptor protein recruitment, and activation of mechanosensing signaling pathways of cells encapsulated in 3D hydrogels.

**Combination of short crosslink lifetime and statically-conjugated ligands promote osteogenic differentiation of encapsulated hMSCs.** We next evaluated the mechanosensing-dependent osteogenesis of hMSCs encapsulated in our hydrogels

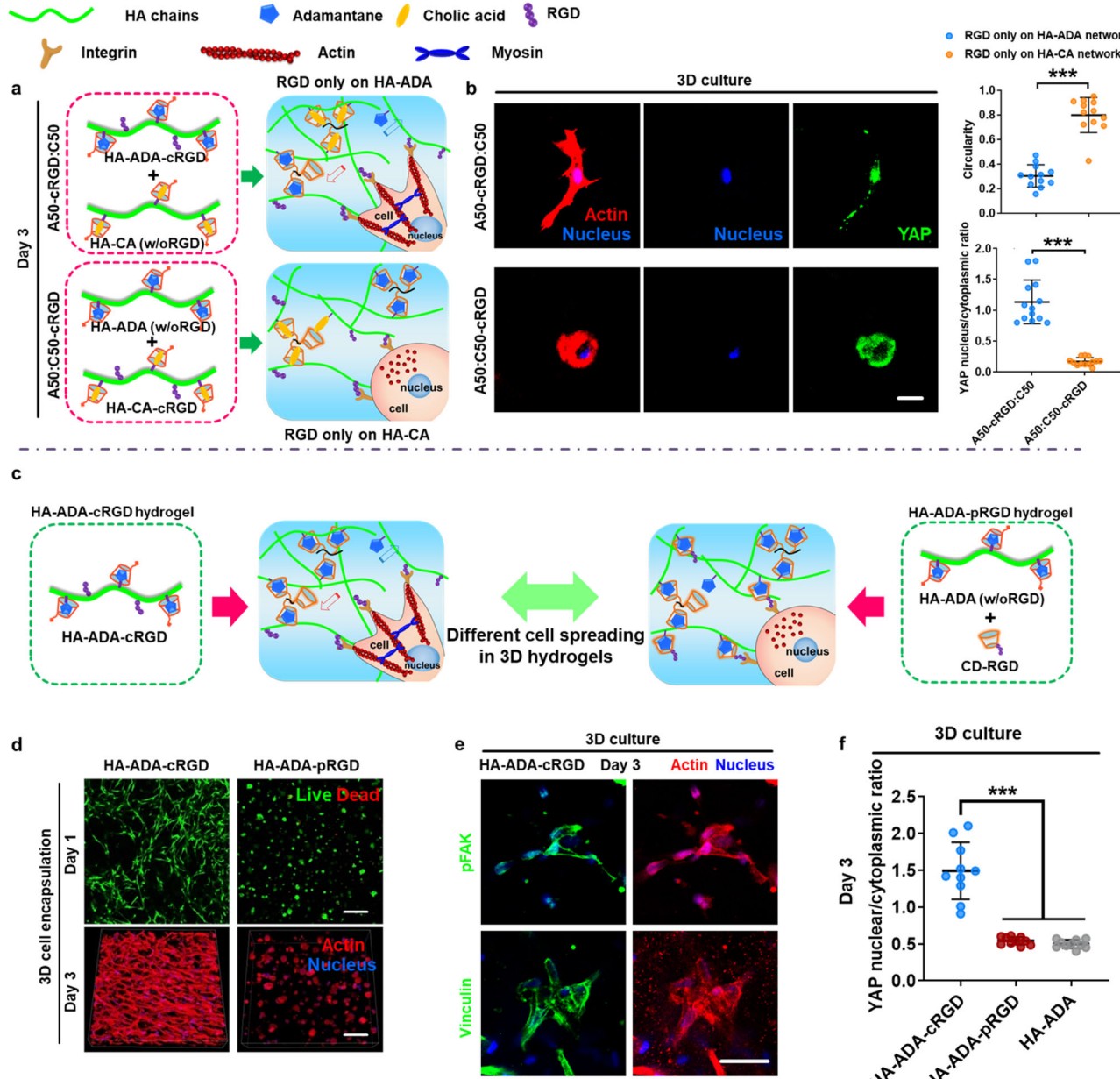

**Fig. 4 The precise and covalent conjugation of cell-adhesive ligands to the hydrogel subnetwork connected by crosslinks with fast binding kinetics is required for efficient 3D spreading and mechanosensing of hMSCs. a** Schematic illustration of the preparation of A50C50 hydrogels with selective RGD conjugations and 3D cell culture in the hydrogels. RGD peptide was only conjugated either to the HA–ADA subnetwork (A50-cRGD:C50) or to HA-CA subnetwork (A50:C50-cRGD) in the A50C50 hydrogels. **b** Representative immunofluorescence staining against F-actin (red), nuclei (blue), and YAP (green) in hMSCs encapsulated in A50C50 hydrogels with selective RGD conjugations after 3 days of culture (scale bar = 25 μm) and quantification of the circularity and nuclear YAP fluorescence intensity (intensity ratio between the nucleus and cytoplasm). Data are presented as mean values ± SD, n = 12 cells per group from two independent hydrogels; ***p < 0.001 (two-tailed Student's t-test). **c** Schematic illustration of the preparation of hydrogels with different RGD conjugation methods (HA–ADA–cRGD and HA–ADA–pRGD) and 3D cell culture with different RGD conjugation methods. **d** Representative images of hMSCs encapsulated within hydrogels (3D cell encapsulation) with different RGD conjugation methods (HA–ADA–cRGD and HA–ADA–pRGD) after 1 and 3 days of culture. Scale bar = 200 μm. **e** Representative immunofluorescence staining against F-actin (red), nuclei (blue) and pFAK or vinculin (green) in hMSCs cultured in highly dynamic HA–ADA–cRGD hydrogels for 3 days. Scale bar = 50 μm. **f** Quantification of the nuclear YAP fluorescence intensity (intensity ratio between the nucleus and cytoplasm) in hMSCs encapsulated in hydrogels with different RGD conjugation methods (HA–ADA–cRGD, HA–ADA–pRGD, and HA–ADA) on after 3 days of culture. Data are presented as mean values ± SD, n = 10 cells per group from two independent hydrogels; ***p < 0.001 (two-tailed Student's t-test).

in vitro. After 7 and 14 days of culture in osteogenic medium, hMSCs in the HA–ADA–cRGD hydrogels exhibit upregulated expression of osteogenic marker genes, including Runx 2, type I collagen, osteocalcin (OCN), and alkaline phosphatase (ALP), compared with cells in the HA–CA–cRGD, HA–ADA–pRGD and HA–ADA hydrogels (Fig. 5a, b and Supplementary Fig. 15). Consistent with the gene expression data, Alizarin red, and Von Kossa staining reveals significant calcification in the HA–ADA–cRGD hydrogels compared with that in the control hydrogels (Fig. 5c). Immunohistochemical staining of histological

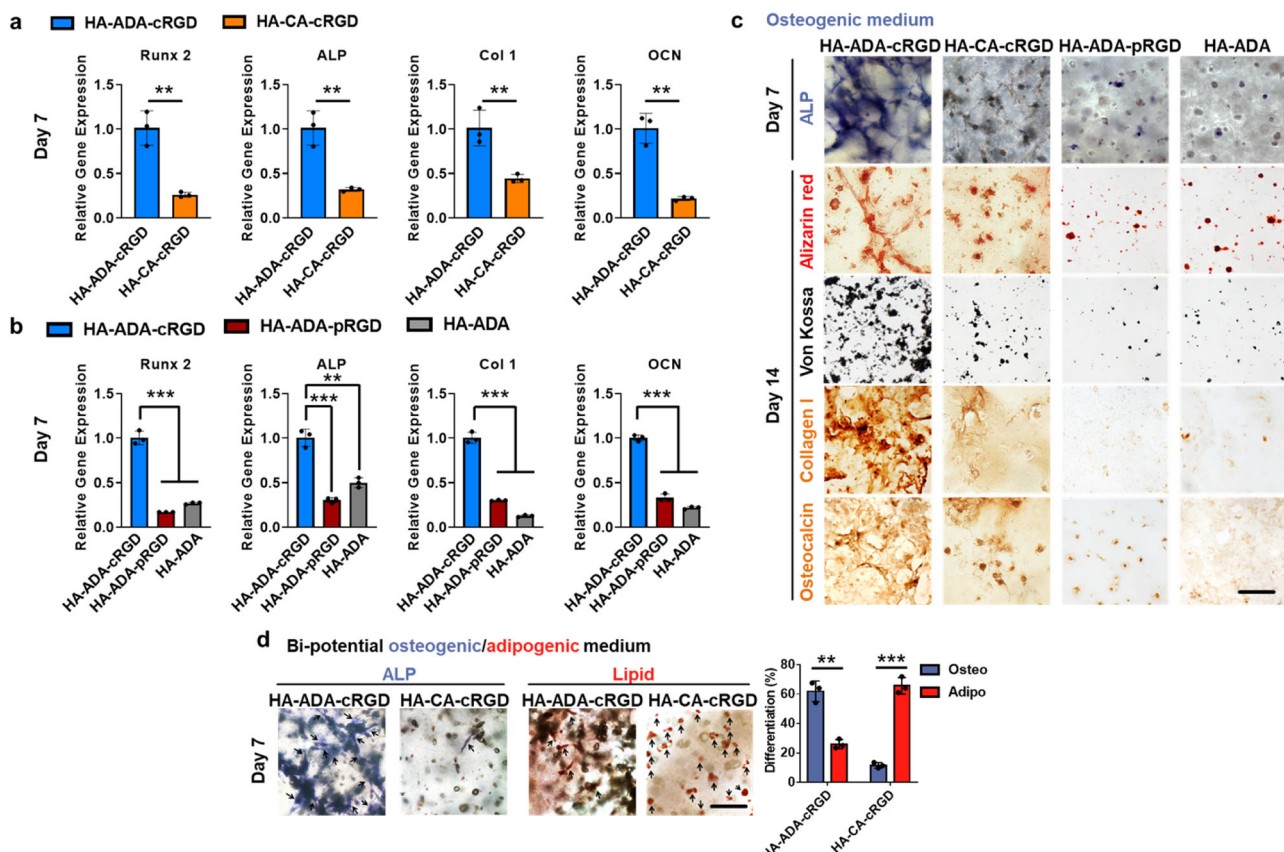

**Fig. 5 Supramolecular hydrogels with short crosslink lifetime promote osteogenic differentiation of encapsulated hMSCs. a** Quantification of Runx 2, ALP, type I collagen, and OCN gene expression of hMSCs encapsulated in the hydrogels (HA–ADA–cRGD and HA–CA–cRGD) by RT-PCR after 7 days of osteogenic culture. Values are normalized to expression levels within HA–ADA–cRGD. Data are presented as mean values ± SD, $n = 3$ independent hydrogels; **$p < 0.01$ (two-tailed Student's $t$-test). **b** Quantification of Runx 2, ALP, type I collagen, and OCN gene expression of hMSCs encapsulated in the hydrogels (HA–ADA–cRGD, HA–ADA–pRGD, and HA–ADA) by RT-PCR after 7 days of osteogenic culture. Values are normalized to expression levels within HA–ADA–cRGD; Data are presented as mean values ± SD, $n = 3$ independent hydrogels; **$p < 0.01$, ***$p < 0.001$ (two-tailed Student's $t$-test). **c** ALP staining (Fast Blue; osteogenic biomarker, blue) of hMSC-laden hydrogels of different groups after 7 days of osteogenic culture, and Alizarin red staining, Von Kossa staining, and type I collagen and OCN immunohistochemical staining of hMSC-laden hydrogels of different groups after 14 days of osteogenic culture (scale bar = 100 μm). **d** Representative ALP and lipid staining and percentage differentiation of hMSCs within HA–ADA–cRGD and HA–CA–cRGD hydrogels following 7 days mixed osteogenic/adipogenic-media incubation (Black arrow pointing to ALP (osteogenesis) or lipid-containing cells) (Scale bar = 100 μm). Data are presented as mean values ± SD, $n = 3$ independent hydrogels; **$p < 0.01$, ***$p < 0.001$ (two-tailed Student's $t$-test).

sections shows the most intense staining against type I collagen and OCN in the HA–ADA–cRGD group. The HA–ADA–cRGD hydrogels also show significantly more ALP staining after 7 days of osteogenic culture (Fig. 5c). Consistently, our Western blot data also show elevated ALP and Runx2 expression in the HA–ADA–cRGD group compared with those in the control groups (Supplementary Fig. 16 and Supplementary Table 4). After 7 days of culture in the presence of bi-potential osteogenic/adipogenic medium, as shown by histological staining for ALP (osteogenesis) and neutral lipids (adipogenesis) (Fig. 5d), the HA–ADA–cRGD hydrogels favor osteogenesis and the HA–CA–cRGD hydrogels favor adipogenesis of the encapsulated hMSCs. These findings indicate that the hydrogels which facilitate cell mechanosensing promote osteogenesis, whereas the hydrogels which hinder cell mechanosensing favor the adipogenesis of the encapsulated hMSCs in 3D culture.

**The molecular basis of the ultra-rapid spreading of hMSCs in ultra-dynamic hydrogels.** We next explored the putative molecular mechanism of ultra-rapid spreading and mechanosensing of hMSCs in the ultra-dynamic HA–ADA–cRGD hydrogels. Given that encapsulated cell behavior within 3D hydrogels can be

regulated by their interactions with nascent proteins deposited by themselves[19,49–51], changes in nascent ECM deposition were also investigated (Fig. 6a). After 18 h in culture, metabolic labeling revealed that cells were surrounded by nascent ECM proteins in HA–ADA–cRGD with a structure that coincided with fibronectin labeling (Fig. 6a). Similarly, hMSCs in HA–CA–cRGD hydrogels deposited ECM proteins (Fig. 6a) but with an increased local ECM thickness (Supplementary Fig. 17), and this is consistent with our previous findings of increased ECM thickness in restrictive hydrogels[19]. Cell spreading was then probed in response to perturbed cellular adhesion to nascent ECM with the addition of a monoclonal antibody that selectively blocks the interaction with the cell-adhesive domain of secreted human fibronectin (HFN 7.1)[19,52] to the culture media. Cell spreading in HA–ADA–cRGD was reduced when the cell interaction with fibronectin was blocked, as indicated by increased circularity after 18 h of culture when compared to untreated controls (Fig. 6b), whereas HFN 7.1 treatment had no effect on cell circularity within HA–CA–cRGD hydrogels (Fig. 6b and Supplementary Fig. 18). The reduction in cell spreading was further dependent on the concentration of added antibody (Fig. 6b and Supplementary Fig. 18), but the addition of antibody did not affect

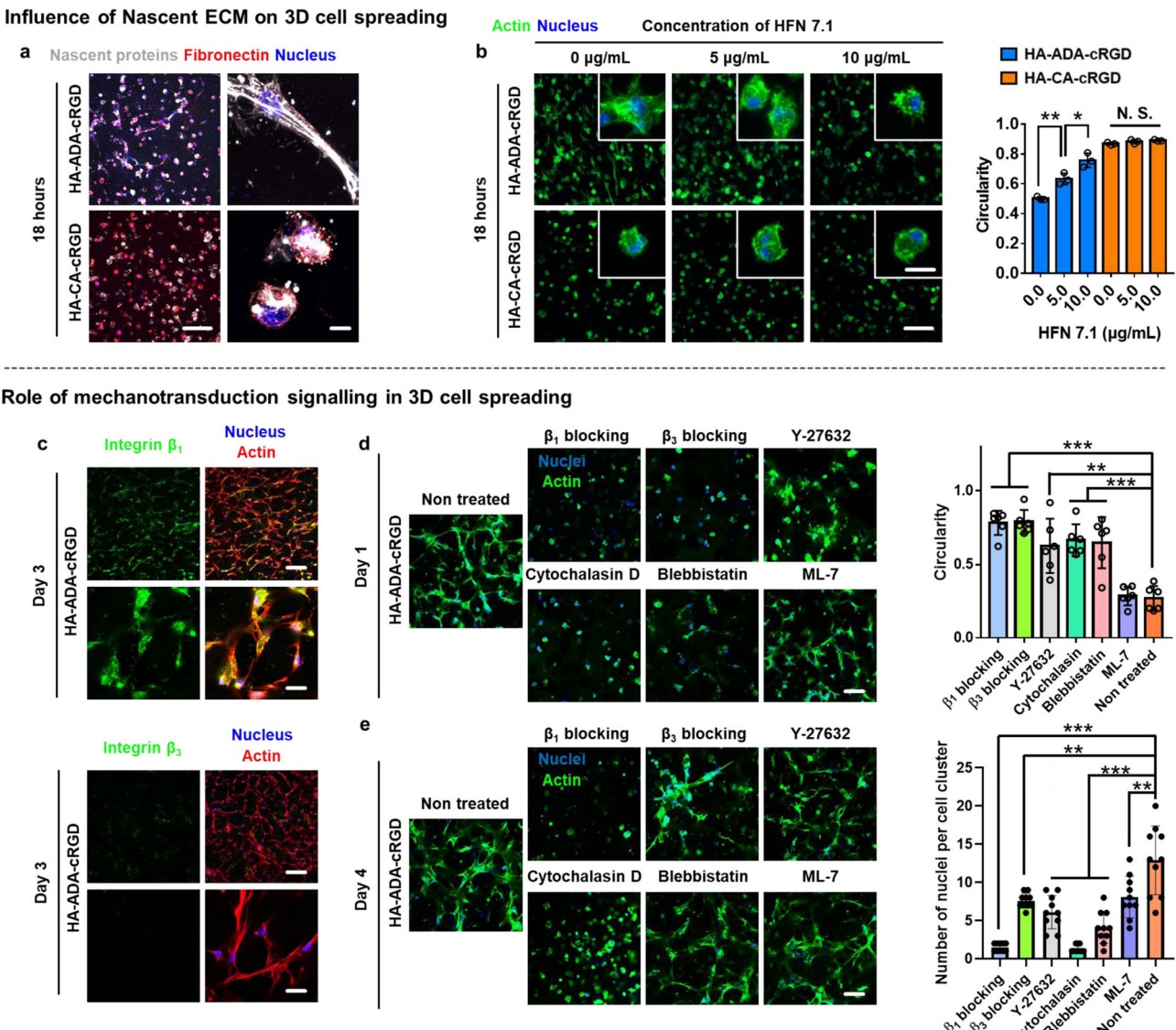

**Fig. 6 The ultra-rapid cell spreading and aggregation in HA–ADA–cRGD hydrogels is regulated by cell–nascent ECM interaction, cell adhesion structures containing β1 class integrins, and actomyosin-based contractility. a** Representative images (magnifications on the right) of nascent proteins (white) and fibronectin (red) secreted by hMSCs encapsulated in HA–ADA–cRGD and HA–CA–cRGD gels within 18 h (scale bars, 200 μm (pictures on the left) and 20 μm (pictures on the right)). **b** Representative images (scale bars, 200 μm (main image) and 10 μm (inset)) of F-actin (green) expressed by the cells encapsulated in HA–ADA–cRGD and HA–CA–cRGD hydrogels after 18 h of treatment with different concentrations of a monoclonal antibody against the cell-adhesive domain of human fibronectin (HFN 7.1, 0, 5, 10 μg/mL, see Supplementary Fig. 19 for cell viability). Circularity of hMSCs encapsulated in HA–ADA–cRGD and HA–CA–cRGD hydrogels after 18 h of treatment with different concentrations of HFN 7.1 (0, 5, 10 μg/mL). Data are presented as mean values ± SD, $n = 3$ independent hydrogels, $*p < 0.05$, $**p < 0.01$ (two-tailed Student's t-test). **c** Representative immunofluorescence staining against F-actin (red), nuclei (blue), and β1 class integrins or β3 class integrins (green) in hMSCs cultured in highly dynamic HA–ADA–cRGD hydrogels for 3 days (images on the top: scale bar = 200 μm. Images on the bottom: scale bar = 50 μm). **d** Cell spreading in the highly dynamic HA–ADA–cRGD hydrogels after 1 day of culture with or without treatment with integrin-blocking antibodies, a myosin inhibitor (blebbistatin), a myosin light chain kinase inhibitor (ML-7), a ROCK inhibitor (Y-27632), or an inhibitor of actin polymerization (Cytochalasin D). Scale bar = 100 μm. The circularity of the hMSCs encapsulated within the hydrogels treated with different inhibitors. (The average circularity value is calculated according to $C = 4\pi A/P^2$, where $A$ is the area occupied by the cell and $P$ is the perimeter of the cell). Data are presented as mean values ± SD, $n = 10$ cells per group from two independent hydrogels; $**p < 0.01$, $***p < 0.001$ (two-tailed Student's t-test). **e** Multi-cell assembly in the highly dynamic HA–ADA–cRGD hydrogels after 4 days of culture with or without treatment with blocking antibodies and inhibitors. Scale bar = 100 μm. The quantification of the multicellularity of the cell clusters within the hydrogels treated with different inhibitors. Data are presented as mean values ± SD, $n = 10$ cell clusters per group from two independent hydrogels; $**p < 0.01$, $***p < 0.001$ (two-tailed Student's t-test).

the cell viability in both HA–ADA–cRGD and HA–CA–cRGD hydrogels (Supplementary Fig. 19).

We further examined the key markers related to the cell–ECM interactions and associated mechanosensing signaling. Our immunofluorescence staining showed the localization of β1 class (e.g.,

α5β1) but not β3 class (e.g., αvβ3) integrins to the periphery of the spreading cells encapsulated in the HA–ADA–cRGD hydrogels, indicating that the maturation of adhesion structures in encapsulated cells is critically dependent on β1 class integrins in our dynamic hydrogels (Fig. 6c). We further quantified the circularity

and multicellularity of the cell clusters (based on the number of nuclei in the cell cluster) of cells encapsulated in HA–ADA–cRGD hydrogels treated with blocking antibodies and pharmacological inhibitors against selected critical components in the mechano-transduction pathways[37]. The non-treated control shows extensive cell spreading and clustering after 1 and 4 days of culture as observed before (Fig. 6d, e). In contrast, inhibition of actin polymerization with cytochalasin D nearly abolishes the stellate spreading (circularity ~1) and assembly of cells in HA–ADA–cRGD hydrogels (Fig. 6d, e), demonstrating the indispensable role of filamentous actin in mediating the ultra-rapid cell spreading. Inhibition of Rho-associated protein kinase (ROCK) with Y-27632, myosin light chain kinase with ML-7, and non-muscle myosin II with blebbistatin reduces the cell spreading and clustering to varying extents (Fig. 6d, e). The cells treated with an antibody against $\beta_1$ class integrins retain the round morphology with only the cortical actin structure in the HA–ADA–cRGD hydrogels, whereas treatment with an antibody against $\beta_3$ class integrins slightly reduces cell spreading on day 1 but did not significantly affect cell spreading on day 4 (Fig. 6d, e). This is consistent with our earlier finding that $\beta_1$ but not $\beta_3$ class integrins are involved in the rapid cell spreading in HA–ADA–cRGD hydrogels.

Together, these findings suggest that the ultra-rapid cell spreading and assembly in the HA–ADA–cRGD hydrogels is mediated by the concerted action of cell adhesion structures containing $\beta_1$ class integrins, interaction with nascent ECM proteins, and actomyosin-based contractility.

## Discussion

In this work, we demonstrate that kinetic constants of physical crosslinks in supramolecular hydrogels are one of the governing factors in determining cell–hydrogel interactions and cell behaviors in 3D. Our experimental and computational findings show that the host–guest crosslinks with large dissociation rate constants and short binding lifetime but not those with small dissociation rate constant and long binding lifetime in the hydrogels facilitate efficient cell spreading and mechanosensing via cell force-mediated network reorganization. Furthermore, the precise and robust conjugation of cell-adhesive motifs to the hydrogel's subnetwork connected by the short binding lifetime crosslinks is essential to the mechanosensing-dependent differentiation of stem cells in 3D, which is based on the formation of cell-adhesive structures and actomyosin contractility. Our supramolecular hydrogels provide a controlled and tunable platform to assist fundamental investigations on cellular responses to dynamic biophysical cues in their 3D microenvironment and an effective delivery vehicle of therapeutic cells for translational studies (Supplementary Fig. 20).

## Methods

**Preparation of HA-ADA and HA-CA hydrogels**. For the supermolecular dynamic hydrogel, functionalized hyaluronic acids (HA–ADA–cRGD or HA–CA–cRGD) were first dissolved in PBS (pH = 7.4) and then mixed with the photoinitiator I2959 (final concentration: 0.05 wt%). The host monomer Ac-β-CD was then added into the solution before it was loaded in homemade molds (13 mm in diameter). After 11 min of UV irradiation (7 mW/cm$^2$), disc-like hydrogels were obtained.

**Cell culture and differentiation**. We acquired passage 4 Normal Human Bone Marrow-Derived Mesenchymal Stem Cells (hMSC) (Lonza, Allendale, New Jersey, USA) expanded in growth medium (α-MEM supplemented with 16.7% FBS, 1% penicillin/streptomycin, and 1% L-glutamine) (Thermo Fisher Scientific, Waltham, Massachusetts, USA). For each hydrogel, the hMSC encapsulation density was 10 million cells/mL. We supplemented all hydrogels with 1 mL of growth media or osteogenic media (10 mM β-glycerophosphate disodium, 50 mg/mL L-ascorbic acid 2-phosphate sesquimagnesium salt hydrate, and 100 nM dexamethasone), which was used for 3D culture in vitro and renewed every 2 days. The samples were collected on days 1, 3, 7, and 14 for the osteogenesis degree evaluation. Mixed

adipogenic/osteogenic inductive media was made by combining osteogenic and adipogenic inductive media in a 1:1 ratio and supplementing with 1% (v/v) penicillin-streptomycin.

**Live/dead assay**. For the cell viability evaluation, cell-laden hydrogels were incubated in PBS with 0.5 μM calcein AM and 4.0 μM ethidium homodimer for 30 min per manufacturer's protocol. After 30 min incubation at 37 °C, the hydrogels were washed three times with PBS, and fluorescent images were acquired with a Nikon C2 + confocal microscope and analyzed using ImageJ (NIH).

**hMSCs spreading**. After encapsulating of hMSCs in the hydrogels, the cells were cultivated in a live-cell incubation chamber. The temperature and atmosphere conditions inside the chamber were controlled to be the same as the cell incubator by a temperature controller. Live-cell time-lapse imaging was conducted using the Nikon C2 + confocal microscope. Cell images were captured with a ×10 objective every 30 min (HA–ADA–cRGD) and 15 min (HA–ADA–pRGD) for a total duration of 24 h.

z-stack images were acquired with 10 × 0.45 NA, 20 × 0.75 NA, and 100 × 1.4 NA by using a Nikon Confocal Microscope. ImageJ 1.50d was utilized to analyze the images. 20 of sections were acquired with 10 × 0.45 NA and 20 × 0.75 NA for the 3D reconstruction.

**Characterization of gel degradation**. For characterization of gel degradation, the hydrogels were formed and incubated in culture media at 37 °C for 1, 3, 7, 14, and 21 days. Then, the hydrogels were removed from the incubation medium, frozen, and lyophilized. The dry mass of the hydrogels was measured following lyophilization.

**Immunostaining**. Cell-laden hydrogels were fixed in 4% paraformaldehyde for 1 h at room temperature, washed with DPBS three times, and then permeabilized with 0.1% (v/v) Triton X-100 in PBS for 30 min at room temperature. Subsequently, constructs were incubated with 5% (w/v) BSA in PBS to block nonspecific binding, after which primary antibodies anti-YAP (mouse monoclonal, Santa Cruz Biotechnology, sc-101199, dilution of 1:200), anti-vinculin (mouse monoclonal, Santa Cruz Biotechnology, sc-25336, dilution of 1:200), anti-integrin (Integrin $\beta_3$, mouse monoclonal, Santa Cruz Biotechnology, sc-71407, dilution of 1:200; Integrin $\beta_1$, mouse monoclonal, Santa Cruz Biotechnology, sc-374429, dilution of 1:200), anti-pFAK (mouse monoclonal, Santa Cruz Biotechnology, sc-374668, dilution of 1:200), anti-fibronectin (mouse monoclonal, Sigma Aldrich, F6140) in 5% (w/v) BSA were added to the constructs and incubated at 4 °C overnight. Hydrogels were then rinsed with 0.5% (v/v) Tween-20 in PBS three times. Secondary antibody goat anti-mouse AlexaFluor® 488 (Invitrogen, dilution of 1:200), phalloidin (Invitrogen, dilution of 1:200), and 4′,6-diamidino-2-phenylindole (DAPI, sigma, 0.1 μg/mL) were added and incubated at room temperature for 90 min. Hydrogels were finally rinsed with 0.5% (v/v) Tween-20 in PBS five times and stored in DPBS before imaging. Fluorescent images were acquired with a Nikon C2 + confocal microscope and analyzed using ImageJ (NIH). Cell clusters with more than two nuclei were defined as multicellular.

**Histological analysis**. The samples were fixed in 4% paraformaldehyde for 24 h, embedded in paraffin, and processed using standard histological procedures. The histological sections (8 mm thick) were stained for targets of interest. Von Kossa stain, ALP stain, and Alizarin red stain were prepared following the manufacturers' instructions. For type I collagen (Col I) and osteocalcin (OCN) immunochemical staining, the sections were stained using a Vectastain ABC kit and DAB Substrate Kit for peroxidase. Briefly, the sections were pre-digested in 0.5 mg/mL hyaluronidase for 30 min at 37 °C and incubated in 0.5 N acetic acid for 4 h at 4 °C to swell the samples prior to overnight incubation with primary antibodies, including mouse monoclonal anti-collagen type I antibody.

**Nascent ECM labeling and antibody inhibition**. Hydrogel constructs were cultured in glutamine, cystine, and methionine-free high glucose DMEM (Life Technologies) supplemented with 10% FBS, 100 μg/mL sodium pyruvate, 0.201 mM cystine, 75 μM azidohomoalanine, 25 μM methionine, 1% penicillin/streptomycin, and osteogenic supplement (R&D System). Blocking of nascent fibronectin adhesion was performed using a monoclonal antibody against human fibronectin (2.5–10 μg/mL HFN7.1, Developmental Studies Hybridoma Bank)[19,53].

For nascent ECM labeling, hydrogels were washed twice in PBS with 2% BSA, followed by 30 min incubation in 30 μM DBCO-488 (Click Chemistry Tools) at 37 °C/5% CO$_2$. Following three washes (3 min each) with PBS/2% BSA, hydrogels were fixed in 10% formalin for 30 min at room temperature and washed with PBS three times. For imaging, z-stack images at 20 × 0.75 NA and 100 × 1.4 NA were acquired using a Nikon A1R Confocal Microscope. Local ECM thickness was determined by creating binary masks of z-stack images in ImageJ with Otsu's thresholding, and the ImageJ plugin "BoneJ" was used to calculate the average local ECM thickness per slice[19,54]. (Nascent ECM thickness: BoneJ Plugin ImageJ (×180, 0.13 μm/pixel), 3–6 slices measured and averaged for each cell)

**Statistics and reproducibility**. GraphPad Prism 7 software was used for all statistical analyses. The data are presented as the mean ± standard deviation. The statistical analysis was performed using two-tailed Student's *t*-tests (for two experimental groups), one-way ANOVA, and Tukey post hoc testing (comparisons among more groups). Three times each experiment was repeated independently with similar results in the Figs. 2a, b, 4d, e, 5c, and 6a, c; and Supplementary Figs. 4, 6, 10, 11a, 12, and 14.

**Reporting summary**. Further information on research design is available in the Nature Research Reporting Summary linked to this article.

## Data availability

All data are available from the authors on reasonable request. Source data are provided with this paper.

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

## Acknowledgements

This work was supported by Group Research Scheme funding from the Chinese University of Hong Kong. The work described in this paper was supported by General Research Funding from the Research Grants Council of Hong Kong (Project Nos. 14204618 and 14205817). The work was partially supported by Hong Kong Research Grants Council Theme-based Research Scheme (Ref. T13-402/17-N). This research is supported by an Innovation Technology Fund (TCFS, GHP/011/17SZ), Hong Kong. We are grateful to our lab members for their assistance.

## Author contributions

L.B., K.W., and B.Y. designed the study, conducted the analysis, and prepared the manuscript; B.G.Y. and K.C.W. synthesized hydrogels and conducted experiments; K.Z., Q.F., R.L., X.X., and S.H.D.W. provided discussion and analyzed the experiment; L.B., Y. W., C.Y., P.Z., C.L., and X.C. contributed to the manuscript writing; Y.W. and C.H.L. designed, conducted, and analyzed the computer modeling; C.L. and J.A.B. investigated the nascent ECM deposition and contributed to the manuscript writing.

## Competing interests

The authors declare no competing interests.
