## [Peer Review File · Nature Communications]

Reviewers' comments:

Reviewer #1 (Remarks to the Author):

Yang and colleagues create supramolecular hyaluronic acid (HA) hydrogels through two related guest-host chemistries with different binding dynamics: a β -cyclodextrin (CD) host complexes with cholic acid (CA) guest with a longer lifetime than it does with adamantane (AD). Though their on-off rate constants differ substantially, the overall equilibrium constants for these reversible reactions are similar - at least as reported for the small molecule interactions in solution (see comments below). This difference is used to create gels of similar G' but tunable G'' . Cellular findings are consistent with others reported in the literature using different gel chemistries, including that mesenchymal stem cells spread more and are pushed towards osteogenic differentiation in gels that undergo faster dynamic rearrangement. Literature has previously described this phenomena using a pericellular "gate" model where gates must open at times matched to cell force exertion on the material, an idea that the authors examine computationally in the submitted article. I found the manuscript to be scientifically sound and the studies to be well executed with appropriate controls and analysis included. Reported findings reinforce concepts already present in the literature with a slightly different material chemistry, but ultimately offer little new biological insight. From this, it seems unlikely that the work will have the scientific impact I would expect from an article published in Nature Communications. Some additional comments are given below:

- The manuscript describes guest-host interactions in terms of forward and reverse complexation reactions with known kinetic constants that have been determined with small molecules. How do these constants and their ratio (equilibrium constant) change when diffusion of the large HA chains is taken into consideration? Along these lines, is it really accurate that both the CD-AD and CD-CA systems have the same number of intact complexes at any time as implied in the paper?
- Physically crosslinked gels are subject to molecular erosion. Does this happen here, and to what extent? Does this vary between gels containing different guest molecules? If so, how much do G' and G'' vary within each system over many days of culture?

Reviewer #2 (Remarks to the Author):

Title.

It is so extensive. Maybe a reduction of the contents will be more clear for the reader: by ex. Dynamic supramolecular hydrogels enhance mechanosensing-dependent development of cells.

Figures.

Figure 1.

In panel a, on the equilibrium directions, from CD-CA; in the $k_{off,ca}$ value is indicating 10 and in the table in b, of the same figure, is indicating $10 E1$. Both mathematical expressions need to be the same.

In panel a, after the indication of the UV effects, is the selection of a single cell in both hidrogels that is represented in the fluorescence microscopical observations? The fluorescence fields are presenting several cells, maybe the selection of a single cell is wrong and the selection needs to be for all the hidrogels in each situation. Or, the fluorescent images are those obtained with the ECM? This is a confusing aspect of this part of the cartoon.

Legend of the figure is not indicating if the shMSC of the panel a, after the UV treatments, are fluorescence stained cells. Also, there is not any indication, at the x-axis of both graphs, for the presence of the A80C20, A50C50 and A20C80 treatments.

Figure 4. panel c, inside of the cartoon where red F-actin is indicated to be attached to RGD, there

are two connecting blue proteins (myosins?) not indicated in the figure legend.

Supplementary fig. 2. Graph C, if similar measurements were performed, the y-axis and x-axis values need to be represented in similar unit formats. Inclusive, panel d needs to be considered too for both graphs

Supplementary fig.6. panel d. In graphs, blue curves are showing $k_{off}=10 \text{ sE-1}$ values, however in the figure legend is indicated 10sE-1 values

Supplementary fig. 8. The green color for the vinculin bar is not clear.

Supplementary fig. 14. Where is the origin of the measurements of the ECM thickness around the hMSCs? Which is the corresponding microscopical image? And how was performed the quantificacion?

Movies

Movies S2, S3 and S4 are not indicated in the manuscript. However, in the PDF format, at lines 234 and 406 there are possible indications as "... (Fig. 2a, Supplementary Fig. 3, Movie S1, 2)..." and "... (Fig. 2a, Supplementary Fig. 3, Movie S3, 4)..." are they ok? Maybe indications as S2 and S4 will be fine. For these movies, It was not possible to find the legends of their contents.

-In the experimental section, there is not any indication how the movies were obtained ...

Results.

-According with the figure 1, panel a, the resultant morphological form of the cells is apparently due the constitution of the hidrogels. However, how sure is the consideration that the conformation of the CD-ADA hidrogels is the responsible of the morphology of the hMSCs and not any possible production of substances after the UV treatment. Could Any possible elimination of these probably produced substances alter the time or the alteration of the morphology of the cells? Inclusive, the initial interaction between the hMSCs and the CD-ADA could be sufficient for inducing some kind of non-stabilization of the cells because of the formation of possible derivates from CD-ADA hidrogels.

-There is not any ultrastructural evidence of the resultant hidrogels in presence or the absence of hMSCs according with figure 1a. Maybe with the help of an AFM will offer more information about the topographical distribution of the hidrogels, the ECM and the adhered cells.

Experimental section.

-hMSC are commercial cells of passage 4. However, there is not more information about the origin of the cells (placental, bone marrow, etc). Because of this, it will be important to search if the origin of the cells could be important for the adaptation of the cells to the hidrogels. Are the authors considering that the same resultant data could be of universal application for all kind of hMSCs?

- -In the immunostaining section, there is not any information about all the utilized conditions for the recovery of images from the confocal microscopy (NA of objectives, type of lasers, emission and detection nm, original software, etc). Also, conditions for analyzing the images from Image J and version of the software are not indicated. Inclusive, for z-stack images, not matter there is indication of the NA, there is no any indication about the number of sections performed and the conditions for performing the 3D reconstruction.

- Fluorescent images need to accompanied with DIC images in order to observe the complete cells without fluorescent staining.

From my point of view, the manuscript is really interesting in order to develop an experimental and synthetic matrix that could resemble to that of the classical ECM produced by hMSC and also, interesting to know how to design the hidrogel in order to have one very dynamic and inducer of the differentiation of the cells. Statistical analyses are clearly showing the reproducibility and reliability of the experiments and they are in accord with the microscopical observations mainly with the support of videos. However, in order to have a major appreciation of the utility of this kind of supramolecular hidrogel for any kind of cells, it will be important to add more information about the used cells or to indicate if there is any other preliminary or published data for application to other similar or different cells. I think that the majority of the view of the present manuscript was intended for showing how the cells are feeling the hidrogels, but there is not any ultrastructural image analysis at this level of the hidrogels, mainly that of the CD-ADA the most

active for the behavior of the cells. Addition of these possible measurements could support better the complete visualization of the formed supramolecular hydrogels, because of that any ultrastructural analysis using SEM, EDS and AFM strategies could offer more view of the importance of these supramolecular hydrogels.

Reviewer #3 (Remarks to the Author):

The manuscript uses HA hydrogels that are crosslinked using dynamic bonds with different dissociation constants but the same equilibrium constant. This results in hydrogels with similar elastic (storage) modulus but different viscous (loss) modulus. The underpinning hypothesis is that different dissociation kinetics in dynamic bonds influences cell spreading, mechanosensing and stem cell differentiation. The idea is certainly interesting but experiments are not conclusive to support this hypothesis. As it is, the paper shows that hydrogels with the same elastic modulus but different viscoelastic properties influence cell mechanosensing. The idea that the rate at which dynamic bonds dissociate is actually the underlying mechanism that leads to hydrogels of different viscoelastic properties has not been demonstrated. Also, the data does not support the idea that the dissociation kinetics has anything to do with cell spreading mechanisms, e.g. actin polymerisation.

Specific comments

1. The paragraph below is speculative as it links dissociation constants with the force that cells perform on the substrate and the binding-unbinding of bonds. It would be relevant to measure the actual dissociation force for the different CD-CA and CD-ADA bonds. This concept is related to the dissociation rate but it is not the same. Why more dynamic crosslinks involve lower dissociation forces? If so, can they be quantified and put in context of cellular traction forces?

"We suspect that for the hydrogel network connected by crosslinks with the long lifetime (i.e., covalent crosslink, CD-CA complexation), the "gates" take too long to open when cells "press" against it by cellular forces and therefore remain closed to hinder cell spreading. In contrast, the pericellular hydrogel network connected by dynamic crosslinks with short lifetime can timely adapt and reorganize in response to cellular forces and may therefore better facilitate the passing through and extension of cell protrusion structures and cell spreading."

2. The idea that cell spreading in 3D involves larger forces and hence larger conjugation affinity of ligands in 3D (compared to 2D) needs to be demonstrated. It would be very useful to have traction force microscopy done in 3D under the different conjugation schemes and compare it to 2D conditions. This would shed some light on whether the physical conjugation of RGD peptide is related to the large-magnitude traction forces. It would also add some quantitative data to this speculative idea which if demonstrated would be of interest.

3. Figure 3 – the data is interesting as seems to support the idea that the actual binding kinetics of host-cell interactions modulates actin polymerisation and then cell spreading. However, Figures 3a-c include calculation for CD-ADA complexation but no simulations have been included for CD-CA and so the data is difficult to assess in the context of the hypothesis of the paper.

4. In relation of Figure 4 and the strength of the RGD interaction with HA chains, authors have shown data just for HA-ADA and CD-RGD. This presumably leads to a highly dynamic (RGD-CD-ADA) link due to the short lifetime of the CD-ADA pair. It would be informative to show similar data for HA-CA-cRGD conjugates where the lifetime of the bond is higher.

5. Vinculin and pFAK images in 3D are rather poor. YAP images are not convincing. In addition to this, there is a long historic debate in the literature about the formation of focal adhesions in 3D. Authors should at least reconsider their results in this context (e.g. Science 200;294(5547):1708-12).

6. Figure 5 shows osteogenic markers in the different hydrogels in osteogenic medium. The data can be read differently: as long as there is proper cell attachment in 3D hydrogels (e.g. bound RGD), osteogenesis happens in osteogenic media. Conversely, reported results do not provide any additional support for the hypothesis investigated in this manuscript, i.e whether having dynamic hydrogels with different dissociation constants (in particular short crosslink lifetime) leads to differential stem cell differentiation. Experiments including HA-CA-cRGD hydrogels and osteogenic media are needed. Also, it would be relevant to investigate whether osteogenic differentiation occurs in HA-CA-cRGD hydrogels in basal media. It was demonstrated that e.g. stiffness of 3D hydrogels modulates stem cell phenotype without changes in cell morphology (Nature Materials 2010;9(6):518-26).

7. The last part of the manuscript is general (Figure 6), with low quality images and it lacks connection to the rest of the manuscript. It is now known that nascent proteins deposited by cells play an important role in 3D hydrogels (e.g. Nature Materials 2019;18:883) but the data included in this manuscript does not provide fundamental insights to contribute to the hypothesis about dynamic hydrogels and dissociation constants. For example, what is the relationship between nascent ECM, focal adhesions and cRGD in these hydrogels. This data does not add anything significantly new to results already presented in the previous mentioned paper.

Reviewers' comments:

Reviewer #1 (Remarks to the Author):

Yang and colleagues create supramolecular hyaluronic acid (HA) hydrogels through two related guest-host chemistries with different binding dynamics: a β -cyclodextrin (CD) host complexes with cholic acid (CA) guest with a longer lifetime than it does with adamantane (AD). Though their on-off rate constants differ substantially, the overall equilibrium constants for these reversible reactions are similar - at least as reported for the small molecule interactions in solution (see comments below). This difference is used to create gels of similar G' but tunable G'' . Cellular findings are consistent with others reported in the literature using different gel chemistries, including that mesenchymal stem cells spread more and are pushed towards osteogenic differentiation in gels that undergo faster dynamic rearrangement. Literature has previously described this phenomena using a pericellular "gate" model where gates must open at times matched to cell force exertion on the material, an idea that the authors examine computationally in the submitted article. I found the manuscript to be scientifically sound and the studies to be well executed with appropriate controls and analysis included. Reported findings reinforce concepts already present in the literature with a slightly different material chemistry, but ultimately offer little new biological insight. From this, it seems unlikely that the work will have the scientific impact I would expect from an article published in Nature Communications. Some additional comments are given below:

Response: We appreciate the reviewer's comments. However, we believe that our work reveals substantial new insights not demonstrated by early works as detailed below.

- a) Our dynamic hydrogels enabled ultra-rapid 3D stellate spreading of the encapsulated hMSCs within 1 day and excessive cell – cell aggregation within 2 – 3 days. Such significant dynamics in hydrogel networks has not been demonstrated in the dynamic hydrogels based on reversible crosslinks previously reported by others including Mooney and Anseth. In contrast to the stellate cell spreading shown in our work, the encapsulated cells only developed some minor cell protrusions while maintaining the largely round cell morphology in these previous reports (Nature Materials, 15, 326 – 334 (2016); Advanced Materials, 26, 865 – 872 (2014)).
- b) We further showed that the dissociation constant (k_{off}) of the reversible crosslinks is the key parameter in mediating the rapid stellate spreading of the encapsulated cells given the similar equilibrium binding constant (K). To the best of our knowledge, although some review papers have described such a notion theoretically, this has not been experimentally verified in the past.
- c) Our dynamic hydrogels are stable during several weeks of culture showing minimal changes in the hydrogel volume due to the unique host-guest macromer (HGM) preassembly approach developed by us (detailed in the following response to Comment No. 2). We believe that our work is the first to demonstrate such ultra-dynamic hydrogel with the combined microscale structural dynamics and bulk scale long-term stability.

- d) We used MD and KMC modelling to further verify the importance of k_{off} in supporting such ultra-rapid stellate cell spreading in the dynamic hydrogels. These calculations show that whether the “gates” blocking the actin polymerization in cells’ filopodia can open fast enough depends crucially on the dissociation rate constants of the host-guest crosslinks. With the same equilibrium constant K_{eq} , the force required and the average time it took for a “gate-opening event” to occur both decreased with increasing k_{off} . Therefore, the hydrogels containing fast crosslinks (CD-ADA) with larger k_{off} but NOT hydrogels containing the slow crosslinks (CD-CA) enabled the rapid cell spreading and mechanosensing activation
- e) More importantly, we also showed that the presentation of the conjugated cell adhesive ligand (RGD peptide) can be significantly influenced by the hydrogel network dynamics. i) the conjugated RGD peptide in the static hydrogel network failed to support cell spreading and mechanosensing, while the conjugated RGD in the dynamic hydrogel can effectively support the ultra-rapid 3D spreading and activated mechanosensing of the encapsulated cells; ii) in the dynamic hydrogels, only the chemically conjugated RGD peptide can provide the required mechanical feedback to support cell spreading and mechanosensing, while the physically conjugated RGD peptide failed to do so; iii) in the hybrid dynamic hydrogel containing both the slow and fast reversible crosslinks, only the precise conjugation of RGD to the subnetwork connected by the fast crosslinks (CD-ADA) but NOT the slow crosslinks (CD-CA) enabled the rapid cell spreading and mechanosensing activation. These interactive effects between the hydrogel network dynamics and the finesse of the ligand conjugation have not been reported before.

The rapid and excessive cell spreading and cell aggregation in our dynamic hydrogels resulted in the significantly accelerated and enhanced mechanotransduction signaling, which is the cornerstone to the initiation of many developmental processes. We believe that our work represents a significant advance in the design of hydrogels with biomimetic biophysical dynamics to effectively support cellular development in 3D.

1. The manuscript describes guest-host interactions in terms of forward and reverse complexation reactions with known kinetic constants that have been determined with small molecules. How do these constants and their ratio (equilibrium constant) change when diffusion of the large HA chains is taken into consideration? Along these lines, is it really accurate that both the C and CD-CA systems have the same number of intact complexes at any time as implied in the paper?

Response: We thank the reviewer for raising this important point. Because CD-AD and CD-CA complexations have similar equilibrium constants, which govern the number of host-guest complexations in the bound state, both HA-ADA and HA-CA hydrogels should contain similar number of intact complexes (bound state). **Based on the equilibrium binding constant K of CD-CA and CD-ADA, the estimation of host-guest complexation rates of both pairs are over 90%.**

Calculation details:

$$K = \frac{[HG]}{[H] \times [G]} = \frac{N \times C}{(C - NC)^2}$$

(2)

So,

$$N = 1 + \frac{1}{2KC} - \frac{1}{2} \times \sqrt{\left(2 + \frac{1}{KC}\right)^2 - 4}$$

(3)

For binding between each β -CD (H) and Guest molecules (G), of which the same bulk concentration was fixed as C, K denotes the binding constant and N denotes the host-guest complexation rates.

Under the main experimental conditions used in this work, the concentrations of β -CD being 0.03 M (same concentration of CA and ADA guest molecules), the host-guest complexation rates of different species could be calculated, and the results are shown in Figure R1.

Figure R1. Host-guest complexation rates of different species.

We fully agree with the reviewer that the binding constants of the host and guest molecules may change when conjugated to polymer chains. However, we note that while the guest molecules (AD or CA) were covalently attached to the HA chains, the host CD molecules were not and can diffuse freely before binding with guest molecules. The impact of conjugation to HA chains on host-guest equilibrium constants, therefore, should be less significant. The accurate determination of such impact is rather challenging requiring very sensitive and sophisticated equipment such as ITC. We are now trying our best to look for such a facility. Meanwhile, because both ADA and CA are conjugated to the HA polymer, we expect such impact on the binding kinetics of ADA and CA with CD will be similar. Therefore, we believe that the conjugation to polymer chains should not substantially compromise our classification of the CD-AD and CD-CA as the fast and slow dynamic crosslinks, respectively.

Reference: *Chem Rev*, 101, 4071-4097 (2001); *Acs Macro Lett*, 2, 278-283 (2013).

2. Physically crosslinked gels are subject to molecular erosion. Does this happen here, and to what extent? Does this vary between gels containing different guest molecules? If so, how much do G' and G'' vary within each system over many days of culture?

Response: We thank the reviewer for highlighting this important issue. As stated in our manuscript, another unique advantage of our dynamic hydrogels is the long-term

stability during cell culture. Traditional supramolecular hydrogels, which are usually formed via the self-assembly of physically interacting biopolymers, are usually weak and less stable than chemical hydrogels. The erosion of these weak physical hydrogels can be significant as indicated by the reviewer.

In contrast, the host–guest macromer (HGM) pre-organization approach we developed here can significantly improve the stability of our host-guest hydrogels. Because the host monomers are less bulky than host polymers, the host–guest complexation is expected to be more efficient than that between host polymers and guest polymers. The photo-polymerization of the acryloyl CD leads to in situ clustering of the preassembled host–guest complexes. The host–guest nanoclusters act as the multivalent cross-linkers of the freestanding HGM hydrogels. Such hydrogels are robust enough to retain the desirable physical properties such as self-healing, compressible, and rapid stress relaxation. We have reported these properties in our previous paper (Macromolecules, 2016, 49, 866–875).

In Supplementary Fig. 5a and b, our degradation test shows that both the cell-laden HA-ADA hydrogels and HA-CA hydrogels have insignificant changes in volume over 14 days of culture. On this basis, we measure the G' and G'' of the cell-laden host-guest hydrogel under the different frequency after 7 days' culture. Compared with the as-prepared hydrogels, G' and G'' did not show a significant change after 7 days of cell culture (Figure R2). These data indicate the good long-term stability and minimal erosion/degradation of the hydrogels during these culture periods.

Figure R2. G' and G'' of the cell-laden host-guest hydrogel in the different frequency after 0 and 7 days culture.

Reviewer #2 (Remarks to the Author):

Title.

It is so extensive. Maybe a reduction of the contents will be more clear for the reader: by ex. Dynamic supramolecular hydrogels enhance mechanosensing-dependent development of cells.

Response: we thank the reviewer for the constructive suggestion. We have shortened the title into “Enhanced mechanosensing of cells in synthetic 3D matrix with controlled biophysical dynamics”.

Figures.

Figure 1.

In panel a, on the equilibrium directions, from CD-CA; in the k_{off} , ca value is indicating 10 and in the table in b, of the same figure, is indicating 10 E1. Both mathematical expressions need to be the same.

Response: we apologize for our lack of consistency. We have changed the number formats and highlighted it in the manuscript with yellow color in Figure 1, Panel a and b.

In panel a, after the indication of the UV effects, is the selection of a single cell in both hydrogels that is represented in the fluorescence microscopical observations? The fluorescence fields are presenting several cells, maybe the selection of a single cell is wrong and the selection needs to be for all the hidrogels in each situation. Or, the fluorescent images are those obtained with the ECM? This is a confusing aspect of this part of the cartoon.

Response: we thank the reviewer for the constructive suggestion. The selection of a single cell was indeed confusing in panel A. We have made changes to this and highlighted it in the manuscript with yellow color in Figure 1, Panel a.

Legend of the figure is not indicating if the hMSC of the panel a, after the UV treatments, are fluorescence stained cells. Also, there is not any indication, at the x-axis of both graphs, for the presence of the A80C20, A50C50 and A20C80 treatments.

Response: In this part, we want to highlight that the cells in the HA-ADA-cRGD hydrogels develop extensive stellate spreading. The image of cells shown in the right part of panel A are the fluorescently stained cells. Upon the reviewer's comments, we have indicated this in the legend to make it easier for readers to understand. A80C20, A50C50 and A20C80 indicate the macromer recipe (e.g., A80C20 indicates 80% HA-ADA + 20% HA-CA) for making hydrogels with different G'' but constant G' . For the presence of the A80C20, A50C50 and A20C80 at the x-axis of both graphs, we have also added the indication to make it easier for readers to follow.

Figure 4. panel c, inside of the cartoon where red F-actin is indicated to be attached to RGD, there are two connecting blue proteins (myosins?) not indicated in the figure legend.

Response: we thank the reviewer for the constructive suggestion. The connecting blue proteins are indeed myosins, and we have added the indication in the figure legend and highlighted it in Figure 4.

Supplementary fig. 2. Graph C, if similar measurements were performed, the y-axis and x-axis values need to be represented in similar unit formats. Inclusive, panel d needs to be considered too for both graphs

Response: we thank the reviewer for the constructive suggestion. We have made corresponding changes to the unit formats and highlighted them in Supplementary Fig. 2.

Supplementary fig.6. panel d. In graphs, blue curves are showing $k_{off}=10 \text{ sE-1}$ values, however in the figure legend is indicated 10sE-1 values

Response: we thank the reviewer for the constructive suggestion. We have made

corresponding adjustments to the data of Supplementary Fig.6.

Supplementary fig. 8. The green color for the vinculin bar is not clear.

Response: we thank the reviewer for the constructive suggestion. In the groups of HA-ADA-pRGD and HA-ADA without RGD, the formation of focal adhesion was suppressed. Therefore, the vinculin staining intensity is reduced due to the suppressed vinculin recruitment in these two groups.

Supplementary fig. 14. Where is the origin of the measurements of the ECM thickness around the hMSCs? Which is the corresponding microscopical image? And how was performed the quantification?

Response: we thank the reviewer for the constructive suggestion. The corresponding representative microscopical images are shown in the Fig. 6a. Regarding the quantification method, we have explained it in the Methods section: Local ECM thickness was determined by creating binary masks of z-stack images in ImageJ with Otsu's thresholding, and the ImageJ plugin "BoneJ" was used to calculate the average local ECM thickness per slice (Nascent ECM thickness: BoneJ Plugin ImageJ (180x, 0.13 $\mu\text{m}/\text{pixel}$), 3-6 slices measured and averaged for each cell).

Reference: *Nat Mater* 18(8):883-891; *Bone* 47(6):1076-1079.

Movies

Movies S2, S3 and S4 are not indicated in the manuscript. However, in the PDF format, at lines 234 and 406 there are possible indications as "... (Fig. 2a, Supplementary Fig. 3, Movie S1, 2)..." and "... (Fig. 2a, Supplementary Fig. 3, Movie S3, 4)..." are they ok? Maybe indications as S2 and S4 will be fine. For these movies, It was not possible to find the legends of their contents.

-In the experimental section, there is not any indication how the movies were obtained ...

Response: we apologize for our negligence. We have changed the indications of the Movie and added the legends for easy reference. In the experimental section, we have added the indication how the movies were obtained.

Results.

-According with the figure 1, panel a, the resultant morphological form of the cells is apparently due to the constitution of the hydrogels. However, how sure is the consideration that the conformation of the CD-ADA hydrogels is the responsible of the morphology of the hMSCs and not any possible production of substances after the UV treatment. Could Any possible elimination of these probably produced substances alter the time or the alteration of the morphology of the cells? Inclusive, the initial interaction between the hMSCs and the CD-ADA could be sufficient for inducing some kind of non-stabilization of the cells because of the formation of possible derivatives from CD-ADA hydrogels.

Response: adamantane and cholic acid are known to maintain their stable chemical structure under UV conditions. In the control group of HA-ADA hydrogel without conjugating the cell adhesive RGD peptide, cells did not show the spreading morphology, even though the control hydrogels were also fabricated with UV-induced photopolymerization. In contrast, after introducing RGD into the HA-ADA hydrogel, obvious cell spreading can be observed. Therefore, we believe that the reason for the rapid spreading of encapsulated cells in the dynamic hydrogels are mainly due to the

dynamic network structure and the conjugated cell adhesive peptide.

-There is not any ultrastructural evidence of the resultant hydrogels in presence or the absence of hMSCs according with figure 1a. Maybe with the help of an AFM will offer more information about the topographical distribution of the hydrogels, the ECM and the adhered cells.

Response: we thank the reviewer for the constructive suggestion. In order to better explain cell spreading and morphology, we used SEM to analyze cell-laden hydrogel. The results showed that hMSCs showed better spreading in the HA-ADA-cRGD hydrogel network, while maintaining a spherical shape in the HA-CA-cRGD hydrogel network. This result is consistent with the fluorescence images of cells observed by confocal microscope (Figure. R3).

Figure R3. SEM images of cell-laden hydrogels. Cells were highlighted with red color.

Experimental section.

-hMSC are commercial cells of passage 4. However, there is not more information about the origin of the cells (placental, bone marrow, etc). Because of this, it will be important to search if the origin of the cells could be important for the adaptation of the cells to the hidrogels. Are the authors considering that the same resultant data could be of universal application for all kind of hMSCs?

Response: we thank the reviewer for the constructive suggestion. The cells that we used here are the passage 4 human bone marrow derived Mesenchymal Stem Cells (hMSC). Now we have added the information in the Supplementary Materials and highlighted it with the yellow color. And for the question “could the same resultant data be of universal application for all kind of hMSCs?” We have begun to study how the hydrogels with different dynamics affect the 3D developments of hMSCs from different tissue origins and hope to report the findings in a follow-up study.

- -In the immunostaining section, there is not any information about all the utilized conditions for the recovery of images from the confocal microscopy (NA of objectives, type of lasers, emission and detection nm, original software, etc). Also, conditions for analyzing the images from Image J and version of the software are not indicated. Inclusive, for z-stack images, not matter there is indication of the NA, there is no any indication about the number of sections performed and the conditions for performing the 3D reconstruction.

Response: we apologize for our negligence. We have added the related info in the revised manuscript as: z-stack images were acquired with 10x0.45 NA, 20x0.75 NA and 100x1.4 NA by using a Nikon Confocal Microscope. Image J 1.50d was utilized to analyze the images. 20 of sections were acquired with 10x0.45 NA and 20x0.75 NA for the 3D reconstruction.

- Fluorescent images need to be accompanied with DIC images in order to observe the complete cells without fluorescent staining.

Response: we thank the reviewer for the constructive suggestion. We have added the IF and DIC image of cells encapsulated in the ADA and CA groups after 1 day culture (Figure R4). The DIC images can help us observe the complete cells without fluorescent staining.

Figure R4. Microscopic images of fluorescently stained hMSCs encapsulated in HA-ADA-cRGD hydrogels after 1 day of culture

From my point of view, the manuscript is really interesting in order to develop an experimental and synthetic matrix that could resemble to that of the classical ECM produced by hMSC and also, interesting to know how to design the hydrogel in order to have one very dynamic and inducer of the differentiation of the cells. Statistical analyses are clearly showing the reproducibility and reliability of the experiments and they are in accord with the microscopical observations mainly with the support of videos. However, in order to have a major appreciation of the utility of this kind of supramolecular hydrogel for any kind of cells, it will be important to add more information about the used cells or to indicate if there is any other preliminary or published data for application to other similar or different cells. I think that the majority of the view of the present manuscript was intended for showing how the cells are feeling the hydrogels, but there is not any ultrastructural image analysis at this level of the hydrogels, mainly that of the CD-ADA the most active for the behavior of the cells. Addition of these possible measurements could support better the complete visualization of the formed supramolecular hydrogels, because of that any ultrastructural analysis using SEM, EDS and ATM strategies could offer more view to the importance of these supramolecular hydrogels.

Response: we appreciate the reviewer's positive comments on our work. We have thoroughly revised our manuscript based on the reviewer's suggestions.

Reviewer #3 (Remarks to the Author):

The manuscript uses HA hydrogels that are crosslinked using dynamic bonds with different dissociation constants but the same equilibrium constant. This results in hydrogels with similar elastic (storage) modulus but different viscous (loss) modulus. The underpinning hypothesis is that different dissociation kinetics in dynamics bonds

influences cell spreading, mechanosensing and stem cell differentiation. The idea is certainly interesting but experiments are not conclusive to support this hypothesis. As it is, the paper shows that hydrogels with the same elastic modulus but different viscoelastic properties influence cell mechanosensing. The idea that the rate at which dynamics bonds dissociate is actually the underlying mechanism that leads to hydrogels of different viscoelastic properties has not been demonstrated. Also, the data does not support the idea that the dissociation kinetics has anything to do with cell spreading mechanisms, e.g. actin polymerisation.

Specific comments

1. The paragraph below is speculative as it links dissociation constants with the force that cells perform on the substrate and the binding-unbinding of bonds. It would be relevant to measure the actual dissociation force for the different CD-CA and CD-ADA bonds. This concept is related to the dissociation rate but it is not the same. Why more dynamic crosslinks involve lower dissociation forces? If so, can they be quantified and put in context of cellular traction forces?

“We suspect that for the hydrogel network connected by crosslinks with the long lifetime (i.e., covalent crosslink, CD-CA complexation), the “gates” take too long to open when cells “press” against it by cellular forces and therefore remain closed to hinder cell spreading. In contrast, the pericellular hydrogel network connected by dynamic crosslinks with short lifetime can timely adapt and reorganize in response to cellular forces and may therefore better facilitate the passing through and extension of cell protrusion structures and cell spreading.”

Response: we thank the reviewer for the insightful comments. More dynamic crosslinks do not necessarily entail lower dissociation forces. On the contrary, after extensive literature search, we now estimate the force required to disconnect CD-CA in the bound state, i.e., its ‘rupture force’, to be ~78 pN, based on an AFM analysis (J. Am. Chem. Soc., 126, 1577 – 1584 (2004)). This is in contrast to the rupture force of 102 pN of CD-ADA. So the more dynamic CD-ADA complexation actually requires a slightly larger force to dissociate in the bound state than the less dynamic CD-CA complexation. However, we observe significantly more cell spreading in the HA-ADA hydrogels (stabilized by the CD-ADA crosslink) than in the HA-CA hydrogels (stabilized by the CD-CA crosslink). This indicates that the cell spreading in the dynamic hydrogels depends more on the dissociation dynamics rather than the dissociation forces of the host-guest crosslinks. This observation is supported by our KMC modeling results, which showed that the required force and the average time it took to “open the gate” blocking actin polymerization were both smaller in HA-ADA than HA-CA (Fig. 3).

Mechanistically, the above result may be explained by the Bell model (Adv Appl Probab, 12, 566-567 (1980); Science, 322, 1687-1691 (2008)), which states that $k_{\text{off}} = k_{\text{off},0} \exp(F/F_0)$, where k_{off} and $k_{\text{off},0}$ are the dissociation rate constants between a host-guest pair with and without the influence of an external force F , respectively, where F_0 represents the rupture force between the given host-guest pair. With typical cellular traction forces ranging from a few piconewtons to on the order of 100 pN, the exponential terms in the above equation should range from ~1 to ~3, with only a small difference between CD-CA and CD-ADA. This is in contrast to the ~100-fold difference in the value of $k_{\text{off},0}$ of these two pairs of host-guest complexations (~10 s⁻¹

for CD–CA and 10^3 s^{-1} for CD–ADA). As a result, with typical cellular traction forces, how fast the two types of host-guest crosslinks unbind, i.e., the value of k_{off} under a given applied force, will be dominated by their intrinsic, base dissociation rate constants, $k_{\text{off},0}$. Consequently, in the dynamic hydrogels containing host-guest crosslinks with higher dissociation rate constant, cellular forces can more easily induce network reorganization and undergo rapid 3D spreading compared with the hydrogels containing host-guest crosslinks with lower dissociation rate constant.

2. The idea that cell spreading in 3D involves larger forces and hence larger conjugation affinity of ligands in 3D (compared to 2D) needs to be demonstrated. It would be very useful to have traction force microscopy done in 3D under the different conjugation schemes and compare it to 2D conditions. This would shed some light on whether the physical conjugation of RGD peptide is related to the large-magnitude traction forces. It would also add some quantitative data to this speculative idea which if demonstrated would be of interest.

Response: we agree with the reviewer that data from the 3D traction force microscopy analysis will provide direct evidence to support our statement that cell spreading in 3D involves larger forces. However, we also realize that the key focus of this work is to examine the impact of the binding kinetics of reversible crosslinks and presentation of bioactive ligands in dynamic hydrogels on cellular developments in 3D matrix. To better emphasize the key message of our work, we have removed the comparison between cell spreading on 2D substrate and in 3D dynamic hydrogels in the revised manuscript. We will dedicate this to a separate study to better answer this important question.

3. Figure 3 – the data is interesting as seems to support the idea that the actual binding kinetics of host-cell interactions modulates actin polymerisation and then cell spreading. However, Figures 3a-c include calculation for CD-ADA complexation but no simulations have been included for CD-CA and so the data is difficult to assess in the context of the hypothesis of the paper.

Response: We thank the reviewer for this important comment, based on which we have revised our manuscript to better reflect the different and yet connected roles of MD and KMC calculations. Briefly, a key output of MD simulation is the measurement of force-induced diffusion of a HA segment with an unbound guest-guest crosslink. While our simulations were only performed for the CD–ADA crosslink, when a host–guest pair is in the unbound state, diffusion of the corresponding HA segment does not depend on the guest (since it is in the unbound state and therefore no host–guest interaction is present) but rather is dominated by the HA chain itself. For this reason, the diffusion coefficient computed from MD simulation, now an input in the subsequent KMC, is used for both CD–ADA and CD–CA crosslinks. As for the MD simulations initiated with CD–ADA in the bound state, the results of MD are in qualitative agreement with an estimation using the Bell model, the latter of which forms the basis of our KMC calculations. The KMC calculations were performed for both CD–CA and CD–ADA. Taken together, the above results suggest that MD simulation of a CD–CA system will not provide significant additional insight into our existing simulation and KMC findings. We have, however, modified both the main text and the SI to reflect the above point and make the distinction as well as the connection between our MD and KMC calculations more clear to the readers.

4. In relation of Figure 4 and the strength of the RGD interaction with HA chains, authors have shown data just for HA-ADA and CD-RGD. This presumably leads to a highly dynamic (RGD-CD-ADA) link due to the short lifetime of the CD-ADA pair. It would be informative to show similar data for HA-CA-cRGD conjugates where the lifetime of the bond is higher.

Response: We thank the reviewer for the constructive comments. We have evaluated the physical conjugation of RGD in the HA-CA hydrogel, and the result showed that similar to the HA-ADA-pRGD hydrogel, cells could not spread in the HA-CA-pRGD hydrogel, either (Figure R5). We conclude that despite the longer bond life time of CA-CD, it is still not stable enough to support the transmission of the traction forces from cells to reorganize the dynamic hydrogel network. More stable conjugation of the cell adhesive RGD peptide such as the covalent conjugation to the hydrogel network is required for effective hydrogel network reorganization by cellular forces.

Figure R5. The morphology of hMSCs encapsulated in HA-CA-pRGD hydrogels after 3 days of culture

5. Vinculin and pFAK images in 3D are rather poor. YAP images are not convincing. In addition to this, there is a long historic debate in the literature about the formation of focal adhesions in 3D. Authors should at least reconsider their results in this context (e.g. Science 200; 294(5547):1708-12).

Response: We apologize for the poor image quality of the Vinculin and pFAK staining. Imaging cells embedded deep in 3D hydrogels is challenging, and our low-end confocal microscope has limited imaging capability. We have redone the immunofluorescence staining of actin, vinculin and pFAK. Actin staining demonstrated a robust network of cytoskeletal structure within cells and vinculin and pFAK staining concentrated at the cell periphery and tips of cell processes in HA-ADA-cRGD hydrogels (Figure R6). The results are similar to that of the previous paper (Nat Mater 12, 458-465 (2013)).

We fully agree with the reviewer that there is a lack of consensus on the formation of FAs in 3D among researchers. In the paper mentioned by the reviewer (Science 2001; 294(5547):1708-12), the matrix which can promote the FA formation in 3D cultures are the gels that present fibrillar architecture. Our HA-ADA-cRGD hydrogels are homogeneous hydrogels and different from the matrix in that paper. To improve the accuracy of our result description, we have replaced the “focal adhesions” with “cell adhesion structures” in the revised manuscript.

Figure R6. Immunofluorescence staining against actin, vinculin, and pFAK in the hMSCs encapsulated in HA-ADA-cRGD hydrogels after 3 days of culture.

For the YAP staining images, we only showed a single cell in each image to better illustrate details of YAP translocation. In fact, we can observe extensive YAP nuclear translocation in the more dynamic hydrogels with precise RGD conjugation (e.g., HA-ADA-cRGD, A50-cRGD:C50). To increase the credibility of YAP nuclear translocation, we have added the YAP staining images with lower magnification, which revealed extensive YAP nuclear translocation (Figure R7). We hope that these images can make the YAP staining results more convincing.

Figure R7. Immunofluorescence staining against YAP in the hMSCs encapsulated in dynamic hydrogels after 1 day of culture.

6. Figure 5 shows osteogenic markers in the different hydrogels in osteogenic medium. The data can be read differently: as long as there is proper cell attachment in 3D hydrogels (e.g. bound RGD), osteogenesis happens in osteogenic media. Conversely, reported results do not provide any additional support for the hypothesis investigated in this manuscript, i.e whether having dynamic hydrogels with different dissociation constants (in particular short crosslink lifetime) leads to differential stem cell differentiation. Experiments including HA-CA-cRGD hydrogels and osteogenic media are needed. Also, it would be relevant to investigate whether osteogenic differentiation occurs in HA-CA-cRGD hydrogels in basal media. It was demonstrated that e.g. stiffness of 3D hydrogels modulates stem cell phenotype without changes in cell morphology (Nature Materials 2010;9(6):518-26).

Response: we thank the reviewer for the constructive comments. Our data presented

in Figure 5a,c,d showed that hMSCs in the HA-ADA-cRGD hydrogels exhibited significantly upregulated expression of osteogenic marker genes, including Runx 2, type I collagen, osteocalcin (OCN), and ALP, compared with cells in the HA-CA-cRGD hydrogels after 7 and 14 days of culture in osteogenic medium. This result showed that even with the chemically conjugated RGD peptide and osteogenic culture medium, the dynamic hydrogel network is still required for the enhanced osteogenesis of encapsulated hMSCs.

Based on the reviewer's suggestion, we also cultured the cell-laden hydrogels in basal medium, and our data show that hMSCs in the HA-ADA-cRGD hydrogels also exhibit upregulated expression of osteogenic marker genes, including Runx 2 and ALP, compared with cells in the HA-CA-cRGD hydrogels after 7 days of culture in basal medium (Figure R8). We agree with the reviewer and the previous finding that the stiffness of the hydrogels itself modulates the cell phenotype without changes in cell morphology. Therefore, we have kept the stiffness of our different hydrogels at similar level, and our major finding is that the fast dissociation reversible crosslinks combined with precise chemical conjugation of cell adhesive ligands in the supramolecular hydrogels network is the key to rapid stellate spreading and enhanced mechanosensing-dependent osteogenesis of encapsulated hMSCs.

Figure R8. Gene expression of osteogenic markers by hMSCs encapsulated in hydrogels after 7 days of culture in basal medium

7. The last part of the manuscript is general (Figure 6), with low quality images and it lacks connection to the rest of the manuscript. It is now known that nascent proteins deposited by cells play an important role in 3D hydrogels (e.g. Nature Materials 2019;18:883) but the data included in this manuscript does not provide fundamental insights to contribute to the hypothesis about dynamics hydrogels and dissociation constants. For example, what is the relationship between nascent ECM, focal adhesions and cRGD in these hydrogels. This data does not add anything significantly new to results already presented in the previous mentioned paper.

Response: we agree with the reviewer that the data presented in Figure 6 are not key findings of our work. Nevertheless, we do think that this part provides additional information to explain our findings presented in the previous sections. The data in Figure 6 showed that cells deposited nascent ECM proteins in different ways in the hydrogels of different dynamics, and the nascent ECM proteins also contribute to the 3D cell spreading in hydrogels. The data presented in Figure 6 also further verified the importance of the classical role of actomyosin-based contractility and mechanotransduction signaling in supporting the rapid 3D spreading of the cells in the

dynamic hydrogels. These data lead to our conclusion that the ultra-rapid cell spreading and assembly in the HA-ADA-cRGD hydrogels is mediated by the concerted action of cell adhesion structures containing $\beta 1$ class integrins, interaction with nascent ECM proteins, and actomyosin-based contractility. We believe that it is necessary to examine the contribution of these previously-reported intracellular and extracellular mechanisms to such ultra-rapid and extensive stellate spreading of hMSCs encapsulated in dynamic hydrogels, which to the best of our knowledge has not been demonstrated in other hydrogels before.

REVIEWER COMMENTS

Reviewer #1 (Remarks to the Author):

I am still somewhat skeptical over whether new biological insight has resulted from these studies. However I do agree with several of the point the authors have identified in their resubmission statement. Since I have no additional scientific concerns at this point, I will support publication if also recommended by the fellow reviewers.

Reviewer #2 (Remarks to the Author):

From my point of view, the new version of the manuscript that is including the answers to my questions and inquiries has been well performed. For me, this new version is including all indicated aspects for considering to be integrated. I have a little concern about this point because the indicated addition is apparently missing in the present new version:

"- Fluorescent images need to be accompanied with DIC images in order to observe the complete cells without fluorescent staining.

Response: We thank the reviewer for the constructive suggestion. We have added the IF and DIC image of cells encapsulated in the ADA and CA groups after 1-day culture (Figure R4). The DIC images can help us observe the complete cells without fluorescent staining."

However, again, if the authors correct this little inconvenience and as I indicated in the previous version, I'm ratifying that the manuscript is really interesting in order to develop an experimental and synthetic matrix that could resemble to that of the classical ECM produced by hMSC and also, interesting to know how to design the hydrogel in order to have one very dynamic and inducer of the differentiation of the cells.

Reviewer #3 (Remarks to the Author):

Authors have improved the manuscript significantly in regards to the original version with additional experimental data that now underpin previously unsupported claims. I will not use this second review of the manuscript to raise new points and I will just comment on my previous concerns:

1. Authors have now obtained literature values for CD-CA and CD-ADA dissociation forces and have concluded that the force needed to dissociate the complexation is not correlated with the dynamics of the bond. While I accept this is a valid point, I still think that there is no demonstration that the dissociation kinetics is the underlying mechanism that controls cell behaviour in these hydrogels and then the novelty of this manuscript.
2. Comparison with spreading in 3D environments has been removed from the revised version of the manuscript.
3. I still think that a manuscript at this level would be stronger by containing both simulations even if authors feel that additional MD simulations of CD-DA systems will not provide additional insights.
4. I feel that there is a contradiction between the outcome of this experiment, i.e. the inability of bound CD-RGD to transmit cellular forces on any of the systems and the underlying hypothesis that the different dynamics of the CD-DA vs CD-ADA pairs – partly triggered by cell forces – determines cell behaviour.

5. The quality of the images is slightly better now.
6. The question about cell differentiation has been addressed.
7. Figure 6 contains observations that do not provide any mechanistic understanding of the role of dynamic hydrogels on cell spreading. What happens with the degradation of the gel as cells spread and how does this alter G'' , i.e. hydrogel dynamics?

Point-by-point response to the reviewers' comments

REVIEWER COMMENTS

Reviewer #1 (Remarks to the Author):

I am still somewhat skeptical over whether new biological insight has resulted from these studies. However, I do agree with several of the points the authors have identified in their resubmission statement. Since I have no additional scientific concerns at this point, I will support publication if also recommended by the fellow reviewers.

Response: We appreciate the reviewer for the support on our work. In fact, we think that the suggestions put forward by the reviewer are very helpful to improving our work.

Reviewer #2 (Remarks to the Author):

From my point of view, the new version of the manuscript that is including the answers to my questions and inquiries has been well performed. For me, this new version is including all indicated aspects for considering to be integrated. I have a little concern about this point because the indicated addition is apparently missing in the present new version: "- Fluorescent images need to be accompanied with DIC images in order to observe the complete cells without fluorescent staining.

Response: We thank the reviewer for the constructive suggestion. We have added the IF and DIC image of cells encapsulated in the HA-ADA and HA-CA groups after 1-day culture (Figure R4). The DIC images can better show the complete cells without fluorescent staining."

However, again, if the authors correct this little inconvenience and as I indicated in the previous version, I'm ratifying that the manuscript is really interesting in order to develop an experimental and synthetic matrix that could resemble to that of the classical ECM produced by hMSC and also, interesting to know how to design the hydrogel in order to have one very dynamic and inducer of the differentiation of the cells.

Response: We appreciate the reviewer's comments. We also agree that DIC images can help better show the complete cells without fluorescent staining. We have added the IF and DIC image of cells encapsulated in the HA-ADA and HA-CA groups at different time points (Day 1 and 7) in the Supporting Information (Supplementary Fig. 6) and highlighted them with yellow colour.

Figure R1. Representative light micrographic image of hMSCs encapsulated in hydrogels from different groups (HA-ADA-cRGD and HA-CA-cRGD) on culture days 1 and 7, actin (red) and nuclei (blue). Scale bar =100 μm .

Reviewer #3 (Remarks to the Author):

Authors have improved the manuscript significantly in regards to the original version with additional experimental data that now underpin previously unsupported claims. I will not use this second review of the manuscript to raise new points and I will just comment on my previous concerns:

1. Authors have now obtained literature values for CD-CA and CD-ADA dissociation forces and have concluded that the force needed to dissociate the complexation is not correlated with the dynamics of the bond. While I accept this is a valid point, I still think that there is no demonstration that the dissociation kinetics is the underlying mechanism that controls cell behaviour in these hydrogels and then the novelty of this manuscript.

Response: We appreciate the reviewer's comment.

In the revised manuscript, we showed that the equilibrium constants of interaction between HA-ADA or HA-CA and CD obtained by *Isothermal Titration Calorimetry* are similar (Supplementary Fig. 2). According to our further simulation results, we estimate that k_{off} for dissociation from the steroid body side of CA is about six orders of magnitude smaller than the ADA dissociation in our system. Therefore, we believe that the dissociation kinetics of the host-guest crosslinks is a key mechanism underlying the drastically different cell behaviours in HA-ADA and HA-CA hydrogels.

In our manuscript, we showed that the dissociation constant (k_{off}) of the reversible crosslinks is the key parameter in mediating the rapid stellate spreading of the encapsulated cells given the similar equilibrium binding constant (K_{eq}). To the best of our knowledge, though some review papers have described such notion theoretically, this has not been experimentally verified in the past. The rapid and excessive cell spreading and cell aggregation in our dynamic hydrogels resulted in the significantly accelerated and enhanced mechanotransduction signalling, which is the cornerstone to the initiation of many developmental processes. We believe that our work represents a significant advance in the design of hydrogels with biomimetic biophysical dynamics to effectively support cellular development in 3D.

2. I still think that a manuscript at this level would be stronger by containing both simulations even if authors feel that additional MD simulations of CD-DA systems will not provide additional insights.

Response: We appreciate the reviewer's comment.

Upon the reviewer's suggestion, we have carried out extensive MD simulations as well as free energy calculations for the CD-CA system as summarized below.

1) Same as in HA-ADA, we performed 240 5-ns pulling simulations with a given CD-CA pair in the free (unbound) state. Analysis of these simulations reveals a similar trend as the HA-ADA system, i.e., when a host-guest pair is in the unbound state, even tens of piconewton forces could induce directed chain movement with free guests and increased force produced increased force-movement correlation, increased speed of HA chain (the bulkier CA results in a slightly smaller diffusion coefficient than ADA), as well as a decreased probability of reunion of unbound host/guest with their original partners. Similarly, results for the corresponding simulations with bound CD-CA are also consistent with results of CD-ADA --- sixty 5-ns runs were performed at the maximum simulated force of 102 pN, which revealed no unbinding event of CD-CA. This result is in line with the CD-CA pair being slower than CD-ADA, the latter of which was found to unbind in two out of sixty runs performed at the same force. These new simulation results have been added to Fig. 3 in the main text as well as Table S2 and Supplementary Fig. 9 in the SI.

2) The above MD simulations also prompted us to launch a series of free energy calculations to better estimate the kinetic constants of CD-CA --- the kinetic constants reported in literatures and used in our previous KMC calculations turned out to be for association/dissociation through the 'thinner' carboxylate end of the CA molecule --- since this end is conjugated to the HA chains in our hydrogel, the association/dissociation of CD-CA need to go through the bulkier steroid body of CA (see the new Supplementary Fig. 8). This will render the slow CA even slower, due to the steric hindrance posed by the steroid body of the molecule. To estimate the corresponding kinetic constants, we performed over 9 microsecond adaptive biasing force (ABF) calculations on the association/dissociation of CD-CA. The resulting free energy landscape is shown in the new Supplementary Fig. 8, which yields good agreement with the experimental reference (*J. Phys. Chem. B*, 123, 9831–9838 (2019)) as well as our own ITC measurements in terms of the location and depth of the energy minimum. As detailed in the updated SI, we used the computed free energy barrier to estimate the k_{off} of CD-CA dissociation through the bulky steroid body to be around 10^{-3} s^{-1} . Considering the potential error in our calculations, we also examined k_{off} values an order of magnitude greater or smaller than the above estimate (see updated Supplementary Fig. 9h). Additionally, we further updated our KMC calculations based on the new MD results from HA-CA --- previously diffusion coefficients obtained from HA-ADA were used in all KMC runs; now diffusion coefficients obtained from HA-CA simulations are used for the corresponding HA-CA KMC calculations. For mixed hydrogels, a weighted average of the two diffusion coefficients based on the mixing ratio is used. Overall, while exact numerical results of the KMC calculations differ, our conclusions remain unaffected --- the probability of achieving 'fast-enough' gate-opening events, i.e., the gate opens within the typical actin polymerization timescale of $\sim 0.01 \text{ s}$, increases with an increasing percentage of the fast CD-ADA crosslinks in the hydrogels. Furthermore, same as before, with the equilibrium binding constant K_{eq} kept unchanged, both the required force and the average time of gate opening in the hydrogels decrease with increasing k_{off} . These new KMC results have been updated in Fig. 3 as well as Supplementary Fig. 9.

Overall, we believe that the above new MD simulations have further strengthened our modeling section and we would like to thank the reviewer for his/her recommendation on simulating the CD-CA system.

3. I feel that there is a contradiction between the outcome of this experiment, i.e. the inability of bound CD-RGD to the transmit cellular forces on any of the systems and the underlying hypothesis that

the different dynamics of the CD-CA vs CD-ADA pairs - partly triggered by cell forces - determines cell behaviour.

Response: We thank reviewer for raising this thoughtful question. As indicated in the manuscript, I believe that both dynamic crosslink consisting of CD-ADA complexation, which allows HA network reorganization by cell forces as also shown by our theoretical modelling, and stable covalent conjugation of RGD to HA chain, which supports effective transmission of cell forces to HA network, are required for efficient 3D spreading of cells in the dynamic hydrogel. Even with the fast dynamic CD-ADA crosslinks in HA-ADA gels, the conjugation of pendant RGD to HA chain via the dynamic CD-ADA complexation is not stable enough to support the effective transmission of cell forces to HA network and associated network reorganization. Vice versa, even with the more stable conjugation of pendant RGD via CD-CA physical interaction, which can improve cell force transmission to HA network, the slow CD-CA crosslinks in the HA-CA hydrogel network still inhibit the 3D cell spreading. In summary, to enable 3D cell spreading, both requirements, i.e., the highly dynamic hydrogel network crosslinks and stable conjugation of cell adhesive ligands to hydrogel network, need to be met at the same time.

4. The quality of the images is slightly better now.

Response: We appreciate the reviewer's comment.

5. The question about cell differentiation has been addressed.

Response: We appreciate the reviewer's comment.

6. Figure 6 contains observations that do not provide any mechanistic understanding of the role of dynamic hydrogels on cell spreading. What happens with the degradation of the gel as cells spread and how does this alter G'' , i.e. hydrogel dynamics?

Response: We appreciate the reviewer's comment.

The data in Figure 6 showed that cells deposited nascent ECM proteins in different ways in the hydrogels of different dynamics, and the nascent ECM proteins also contribute to the 3D cell spreading in hydrogels. The data presented in Figure 6 also further verified the importance of the classical role of actomyosin-based contractility and mechanotransduction signalling in supporting the rapid 3D spreading of the cells in the dynamic hydrogels. The data presented in Figure 6 lead to our conclusion that the ultra-rapid cell spreading and assembly in the HA-ADA-cRGD hydrogels is mediated by the concerted action of cell adhesion structures containing β_1 class integrins, interaction with nascent ECM proteins, and actomyosin-based contractility. We think that the data in Figure 6 are helpful to clarify the mechanism of cell spreading in a three-dimensional dynamic hydrogel.

Regarding the effects of gel degradation as cells spread, we thank the reviewer for highlighting this important issue. As we responded to the reviewer #1 before (Question 2), our dynamic hydrogels exhibit good long-term stability during cell culture. We measure the G' and G'' of the cell-laden host-guest hydrogel under the different frequencies after 7 days of culture. Compared with the moduli of as prepared hydrogels, the G' and G'' did not show a significant change after 7 days of cell culture (Figure R2). The main purpose of this manuscript is to illustrate how the hydrogel network affects cell behaviour in the early time point in vitro. The degradation of the dynamic hydrogel in vivo and its impact on cell behaviour is indeed an interesting topic, and we plan to investigate this issue in our future research. Thanks again to the reviewer for this valuable comment.

Freshly prepared cell-laden HA-ADA hydrogel (Day 0)

Cell-laden HA-ADA hydrogel after 7 days of culture

Freshly prepared cell-laden HA-CA hydrogel (Day 0)

Cell-laden HA-CA hydrogel after 7 days of culture

Figure R2. G' and G'' of the freshly prepared and cultured (7 days) cell-laden HA-ADA hydrogels.

REVIEWER COMMENTS

Reviewer #3 (Remarks to the Author):

I am pleased that authors have put additional work to address my original concerns. I believe that the manuscript has been sufficiently improved now.

Manuel Salmeron-Sanchez